# Global Optimal K-Medoids Clustering of One Million Samples

**Jiayang Ren**[†]**, Kaixun Hua**[†]**, Yankai Cao**[*]
University of British Columbia

## Abstract

We study the deterministic global optimization of the K-Medoids clustering problem. This work proposes a branch and bound (BB) scheme, in which a tailored Lagrangian relaxation method proposed in the 1970s is used to provide a lower bound at each BB node. The lower bounding method already guarantees the maximum gap at the root node. A closed-form solution to the lower bound can be derived analytically without explicitly solving any optimization problems, and its computation can be easily parallelized. Moreover, with this lower bounding method, finite convergence to the global optimal solution can be guaranteed by branching only on the regions of medoids. We also present several tailored bound tightening techniques to reduce the search space and computational cost. Extensive computational studies on 28 machine learning datasets demonstrate that our algorithm can provide a provable global optimal solution with an optimality gap of 0.1% within 4 hours on datasets with up to one million samples. Besides, our algorithm can obtain better or equal objective values than the heuristic method. A theoretical proof of global convergence for our algorithm is also presented.

## 1 Introduction

In this paper, we concentrate on the K-Medoids clustering problem: given a dataset, K-Medoids aims to select $K$ samples from the dataset as cluster medoids that minimize the sum of dissimilarities from other samples to the closet medoids. Here, selecting from existing samples as medoids is called the "medoids on samples" constraint. K-Medoids is similar to the K-Means problem except for the arbitrary dissimilarity measurements and the "medoids on samples" constraint. Hence, K-Medoids generally has better interpretability than K-Means [1]. K-Medoids problems are generally NP-hard to solve exactly. So there are many heuristic methods, such as PAM [2], CLARA [3], K-means-like method [4], and Fast-PAM [5]. However, none of them can guarantee reaching the global minimum.

To solve the K-Medoids problem deterministically, back in the 1970s, Cornuejols et al. [6] proposed a Lagrangian relaxation method for obtaining a lower bound of the optimal value, which can guarantee a relative optimal gap smaller than $1/e$. Interestingly, to the best of our knowledge, this is the only application in which the Lagrangian duality gap can be guaranteed within a certain threshold. Several improvements to this method are proposed in the literature. One common direction is improving the heuristic process of Lagrangian dual problems. [7] constructed a semi-Lagrangian relaxation, then accelerated the heuristic process by a cutting plane method. It was further improved in [8] by fixing variables in the heuristic process. Their works can deal with medium-scale datasets (e.g., 3000 samples). Another common direction is column generation [9, 10].

One limitation of these Lagrangian relaxation based methods is that there is no guarantee of convergence to the global optimal solution, due to the existence of duality gap. To address the global

---

[*]corresponding author: yankai.cao@ubc.ca.

[†]Jiayang Ren (rjy12307@mail.ubc.ca) and Kaixun Hua (kaixun.hua@ubc.ca) contributed equally.

36th Conference on Neural Information Processing Systems (NeurIPS 2022).

optimum guarantee, a few works have utilized the branch and bound (BB) scheme in K-Medoids [11, 12, 13]. Christofides and Beasley [11] proposed a prototype by branching on binary variables and was capable of small-scale datasets (200 samples). This scheme was further improved by [12] through branching on semi-assignment constraints, which obtained solutions faster on small-scale datasets. [13] presented a tighter form of the K-Medoids problem and was capable of medium-scale datasets (900 samples). Besides these specific works, several off-the-shelf solvers also contain well-implemented BB schemes, such as CPLEX [14] and Gurobi [15]. These solvers can typically address the K-Mediods problem with medium-scale datasets (e.g., 2000 samples).

The BB schemes used in the aforementioned works can only address medium-scale datasets because they need to branch on all the binary variables to guarantee convergence, which leads to poor scalability because the number of binary variables increases linearly as the number of samples. Therefore, we focus on the reduced-space BB scheme proposed in the stochastic programming community, which only needs to branch on the space of coupling variables (e.g., medoids). This idea was first reported for stochastic mixed-integer linear programs in [16] and then extended to stochastic mixed-integer nonlinear programs in [17]. The convergence for stochastic nonlinear programs was proved in [18]. Recently, the authors in [19] adopted this method and reported the capacity of solving the K-means clustering problem (a similar clustering task without the "medoids on samples" constraint) with 210,000 samples in parallel to an optimal gap of 2.5%. However, the convergence of the reduced-space BB scheme can only be guaranteed when it is combined with suitable lower bounding methods. In previous works, lower bounds are constructed by the Lagrangian relaxation [16] [17] [19] or removal [18] of the so-called non-anticipativity constraints (an implicit constraint that all the samples share the same coupling variables, that is Constraint 10 in this paper). However, these methods are unsuitable for the K-Medoids problem, since they cannot address the "medoids on samples" constraint (a constraint that has a large dimension and connects all samples) and thus cannot ensure the convergence of the reduced-space BB scheme. Moreover, these lower bounding methods require the solution of many expensive mixed-integer nonlinear programs to global optimality at each node. In contrast, the work of Cornuejols et al. [6] generates lower bounds for the K-Medoids problem by the Lagrangian relaxation of the semi-assignment constraint (Constraint 2b in this paper), and the formulation directly takes the "medoids on samples" constraint into consideration. This method has a closed-form solution, and its computation can be easily parallelized.

**Our Contributions:** Firstly, we combine the method of reduced-space BB scheme and the work of Cornuejols et al. [6], and adapt these methods for the K-Medoids problem. The Lagrangian relaxation in [6] is used to generate a tight lower bound, which already guarantees a maximum gap at the root node. With this lower bound, we can prove the finite convergence (detailed proofs presented in the Appendix A) by branching only on the regions of medoids (the dimension of which is independent of the number of samples). Secondly, several tailored bound tightening techniques, including probing and feasibility based methods, are proposed to significantly reduce the search space and computational cost of each node. Lastly, we propose an upper bounding method by heuristic updates of the candidate solutions obtained in the lower bounding procedure. These contributions enable our algorithm to be extremely scalable.

**Capability For Million Scale Problems:** we provide an open-source project in Julia and present numerical experiments on 28 datasets. Specifically, for two datasets with one million samples (which is **400 times larger** than the state-of-art BB method), our algorithm can reach a small gap (0.1%) within one hour. To the best of our knowledge, no other article has ever reported results on datasets of this scale for K-Medoids problems with a global convergence guarantee.

## 2 Preliminaries

**K-Medoids problem:** we denote a dataset with $S$ samples and $A$ attributes as $X = \{x_1, ..., x_s, ..., x_S\} \in \mathbb{R}^{S \times A}$, in which $x_s = [x_{s,1}, ..., x_{s,A}] \in \mathbb{R}^A$ is the $s$th sample, and $x_{s,a}$ is the $a$th attribute of $s$th sample. The K-Medoids problem aims to select a subset from the dataset as the cluster medoids with the following objective:

$$\min_{\mu \in \overset{\circ}{M} \cap X} \sum_{s \in \mathcal{S}} \min_{k \in \mathcal{K}} ||x_s - \mu^k||_2^2, \tag{1}$$

where $\mu^k := [\mu_1^k, ..., \mu_A^k] \in \mathbb{R}^A$ is the medoid of $k$th cluster, $\mu := \{\mu^1, ..., \mu^K\}$ is the medoids set, $\mathcal{S} := \{1, ..., S\}$ is the index set of samples in the dataset, $\mathcal{K} := \{1, ..., K\}$ is the index set of clusters,

$\mathcal{A} := \{1, ..., A\}$ is the index set of attributes. We use $\mu \in X$ to represent the "medoids on samples" constraint in the K-Medoids problem. To facilitate the discussion of the BB scheme, we introduce $\overset{\circ}{M} := \{\mu | \overset{\circ}{\underline{\mu}} \le \mu \le \overset{\circ}{\bar{\mu}}\}$ as the initial regions of medoids. Here, $\overset{\circ}{\underline{\mu}}$ and $\overset{\circ}{\bar{\mu}}$ are the lower and upper bounds of medoids referred from data. Specifically, $\overset{\circ}{\underline{\mu}}_a = \min\limits_{s \in S} x_{s,a}$ and $\overset{\circ}{\bar{\mu}}_a = \max\limits_{s \in S} x_{s,a}$ are the bounds of the $a$th attribute. Note that the introduction of $\overset{\circ}{M}$ does not affect the optimal solution.

Equivalently, the K-Medoids problem can also be expressed in an extensive form (EF) [20]:

$$\text{(EF)} \quad z(\overset{\circ}{M}) = \min_{b,y} \sum_{s \in \mathcal{S}} \sum_{j \in \mathcal{S}} d_{s,j} b_{s,j} \tag{2a}$$

s.t.

$$\sum_{j \in \mathcal{S}} b_{s,j} = 1 \quad \text{(2b)}$$

$$\sum_{j \in \mathcal{S}} y_j = K \quad \text{(2c)}$$

$$b_{s,j} \le y_j \tag{2d}$$

$$b_{s,j}, y_j \in \{0, 1\} \tag{2e}$$

$$s, j \in \mathcal{S}, \ k \in \mathcal{K}, \tag{2f}$$

where $d_{s,j} = ||x_s - x_j||_2^2$ is the distance between samples $x_s$ and $x_j$; $y_j$ is equal to 1 if sample $x_j$ is a medoid, and otherwise 0; $b_{s,j}$ is equal to 1 if sample $x_s$ belongs to the cluster whose medoid is $x_j$, and otherwise 0. Note here $d_{s,j}$ is computed offline, so arbitrary dissimilarity measurements can be used without affecting the solution process. Constraint 2b ensures that each sample $x_s$ must belong to one cluster. Constraint 2c guarantees that a total of $K$ samples are selected as medoids. Constraint 2d expresses that sample $x_s$ can be assigned to sample $x_j$ only if $x_j$ is the medoid of a cluster.

**Lagrangian Relaxation of the K-Medoids problem:** by the Lagrangian relaxation of the Constraint 2b, we can have a relaxed problem of the Extensive Form 2:

$$\beta_{LD}(\overset{\circ}{M}, \lambda) = \min_{b,y} \{\sum_{s \in S} [\sum_{j \in S} (d_{s,j} - \lambda_s) b_{s,j} + \lambda_s]\} \tag{3}$$

s.t. Constraints in the Extensive Form 2 except 2b,

where $\lambda_s$ is the Lagrangian multiplier for $s$th sample. We define the multiplier set as $\lambda := \{\lambda_1, ..., \lambda_S\}$. For a given $\lambda$, $\beta_{LD}(\overset{\circ}{M}, \lambda)$ can be solved analytically. Considering the objective function and constraints of the Relaxed Problem 3, we note that the optimal values for $b_{s,j}$ are given by

$$b_{s,j} = \begin{cases} y_j, & \text{if } d_{s,j} - \lambda_s \le 0, \\ 0, & \text{otherwise.} \end{cases} \tag{4}$$

We define the contribution of $j$th sample to the objective function as:

$$\rho_j(\lambda) := \sum_{s \in S} \min(0, d_{s,j} - \lambda_s). \tag{5}$$

The Lagrangian problem can be reduced to

$$\beta_{LD}(\overset{\circ}{M}, \lambda) = \min_{y_j \in \{0,1\}, j \in \mathcal{S}} \{\sum_{j \in S} \rho_j(\lambda) y_j + \sum_{s \in S} \lambda_s\}$$

$$\text{s.t.} \quad \sum_{j \in \mathcal{S}} y_j = K \tag{6}$$

Its closed-form solution can be easily derived by selecting $y_j$ corresponding to the K smallest $\rho_j$.

**Lagrangian dual:** a proper selection of $\lambda$ may generate a tighter lower bound. Hence, we need to solve the Lagrangian dual problem to get the tightest lower bound:

$$\beta_{LD}(\overset{\circ}{M}) = \max_{\lambda} \beta_{LD}(\overset{\circ}{M}, \lambda). \tag{7}$$

There are several off-the-shelf heuristic methods to update $\lambda$, such as Sub-gradient method [11], Volume method [21] and Cutting-Plane method [22]. The solution to the Dual Problem 7 provides a lower bound to the original problem, with $\beta_{LD}(\overset{\circ}{M}) \le z(\overset{\circ}{M})$. Particularly, for this problem, there is a unique guarantee of maximum gaps between the lower bounds and the global optimal value, which is $\frac{z(M) - \beta_{LD}(\overset{\circ}{M})}{z_r - \beta_{LD}(\overset{\circ}{M})} < \frac{1}{e}$, $z_r = \sum\limits_{s \in \mathcal{S}} \max\limits_{j \in \mathcal{S}} d_{s,j}$. This guarantee is theoretically proved in [6, 23].

# 3 Lagrangian Based Lower Bound

The aforementioned Lagrangian method cannot close the duality gap. To obtain a global optimal solution, we propose to combine it with the spatial branch and bound (BB) scheme, in which Lagrangian relaxation provides a lower bound for each BB node.

**Tailored Lagrangian Relaxation in BB nodes:** Since our BB scheme branches on the regions of the medoids $\mu$, at each BB node, we solve the problem with the partition set $M \subseteq \overset{\circ}{M}$. That is, a constraint $\mu \in M$ is added at each BB node. We further define $M^k := \{\mu^k | \underline{\mu}^k \leq \mu^k \leq \bar{\mu}^k\}$ as the medoid region of the $k$th cluster, where $\underline{\mu}^k$ and $\bar{\mu}^k$ are the lower and upper bounds of the $k$th medoids. Since the possible region of each cluster's medoid can be different, we need to know the assignment of medoids to each cluster. Hence, we replace $y_j$ with $y_j^k$ in EF 2: $y_j^k$ is equal to 1 if sample $x_j$ is selected as the medoid of the $k$th cluster, and otherwise 0. The Constraint 2d is replaced by

$$\sum_{j \in \mathcal{S}} y_j = K \quad \text{(2d)} \quad \implies \quad \sum_{j \in \mathcal{S}} y_j^k = 1 \quad \text{(2d.1)}, \quad \sum_{k \in \mathcal{K}} y_j^k \leq 1 \quad \text{(2d.2)}.$$

Notice that $\mu^k$ is not directly modelled in EF, we need to transform the medoid regions $\mu^k \in M_k$ to the constraints of $y_j^k$. The regions of medoids implicitly control the selection of $y_j^k$ because we have $y_j^k = 0$, if $x_j \notin M^k$. Hence, we define the feasible index set of $k$th cluster's medoid as $\mathcal{S}^{k+}(M) := \{j \in \mathcal{S} \mid \underline{\mu}^k \leq x_j \leq \bar{\mu}^k\}$. In this way, we can reformulate the reduced Lagrangian Problem 6 at the BB node with $M$ as follow:

$$\beta_{LD}(M, \lambda) = \min_{y_j^k \in \{0,1\}, k \in \mathcal{K}, j \in \mathcal{S}^{k+}(M)} \{\sum_{k \in \mathcal{K}} \sum_{j \in \mathcal{S}^{k+}(M)} \rho_j(\lambda) y_j^k + \sum_{s \in \mathcal{S}} \lambda_s\} \tag{8}$$

$$\text{s.t.} \quad \text{Constraint (2d.1), (2d.2).}$$

with the Lagrangian dual problem defined as $\beta_{LD}(M) = \max_\lambda \beta_{LD}(M, \lambda)$. The optimal value of $y_j^k$ in Problem 8 can be obtained by the following procedure: (1) initialize $k = 0$ and $J^* = \emptyset$ (2) let $j^{k*}$ be the index of smallest $\rho_j(\lambda)$ subject to $j \in \mathcal{S}^{k+}(M)$ and $j \notin J^*$; (3) set $y_j^k = 1$ if $j = j^{k*}$, otherwise, 0; (4) add $j^{k*}$ to $J^*$; (5) update $k = k + 1$ and return to (2) until $k = K$.

**Parallel implementation:** we implement a parallel version of the tailored Lagrangian relaxation, in which the computations of contributions, $\rho_j(\lambda)$, are performed in each process. At the beginning of the BB procedure, given $P$ processes, the index set $\mathcal{S}$ of the whole dataset is evenly divided into $P$ subsets, $\mathcal{S}_p, p = 1, .., P$. Then, each process computes and stores the distance assigned to it, which is $d_{s,j}, s \in \mathcal{S}, j \in \mathcal{S}_p$. At each BB node with $M$, denote the feasible index set of all the medoids as

$$\mathcal{S}^+(M) = \mathcal{S}^{1+}(M) \cup \mathcal{S}^{2+}(M) \cup ... \cup \mathcal{S}^{K+}(M). \tag{9}$$

Then, each process computes the assigned contributions using the stored $d_{s,j}$, which is $\rho_j(\lambda), \forall j \in \mathcal{S}_p \cap \mathcal{S}^+(M)$. The remaining parts of the parallel implementation (e.g., update of $\lambda$) are identical to the serial version.

# 4 Tailored Reduced-space Branch and Bound Scheme

This section introduces the tailored reduced-space branch and bound algorithm for the K-Medoids problem, which is detailed in Algorithm 1. Specifically, starting from the root node with $\overset{\circ}{M}$, the BB scheme recursively partitions the medoid regions. At each iteration, the algorithm first selects the node with the lowest lower bound and denotes it as $M$. Then, bound tightening techniques are applied to get a tightened medoid region $\hat{M}$. The lower and upper bounds of the tightened node are computed and updated sequentially. If the gap is larger than the tolerance, we branch on the medoid variable $\mu_a^k$ with the maximum range to get two sub-nodes $M_1$ and $M_2$ with $relint(M_1) \cap relint(M_2) = \emptyset$, where $relint(M)$ is the relative interior of $M$. On the convergence of the algorithm, we have the following theorem with detailed proofs in Appendix A:

**Theorem 1.** *Algorithm 1 is convergent to the global optimal solution after a finite step L, with $\beta_L = z = \alpha_L$, by only branching on the space of $\mu$.*

**Algorithm 1** Tailored Reduced-space Branch and Bound Scheme for K-Medoids clustering

1: **Initialization**
2: Load and store dataset as $X$.
3: Initialize the iteration index $l = 0$, the node set $\mathbb{M} \leftarrow \overset{\circ}{M}$, and tolerance $\epsilon \geq 0$.
4: Compute initial lower, upper bounds: $\beta_l = \beta(\overset{\circ}{M})$, $\alpha_l = \alpha(\overset{\circ}{M})$.
5: **while** $\mathbb{M} \neq \emptyset$ and $|\beta_l - \alpha_l| \geq \epsilon$ **do**
6:    **Node Selection**
7:    Select and delete from $\mathbb{M}$ a Node $M \in \mathbb{M}$ with its parent node's lower bound $\beta(M_P) = \beta_l$ (for the root node, select and delete itself).
8:    Update $l \leftarrow l + 1$.
9:    **Bound Tightening**
10:    Probing // Alg. 2.
11:    Cluster Assignment // Alg. 3.
12:    Feasibility Based Bound Tightening // Alg. 4.
13:    Tighten according to the "medoids on samples" constraint.
14:    Obtain the tightened node $\hat{M}$.
15:    **Bounding**
16:    Compute lower, upper bounds: $\beta(\hat{M})$, $\alpha(\hat{M})$.
17:    Update $\beta_l \leftarrow min\{\beta(M') \mid M' \in \mathbb{M}\}$.
18:    Update $\alpha_l \leftarrow min\{\alpha_{l-1}, \alpha(\hat{M})\}$.
19:    Delete all $M'$ from $\mathbb{M}$ if $\beta(M') > \alpha_l$.
20:    **Branching**
21:    **if** $|\hat{M} \cap X| > 1$ and $|\beta_l - \alpha_l| \geq \epsilon$ **then**
22:      Partition $\hat{M}$ into subsets $M_1$ and $M_2$ with $relint(M_1) \cap relint(M_2) = \emptyset$.
23:      Add $M_i$ to $\mathbb{M}$ if $M_i^k \cap X \neq \emptyset, \forall k \in \mathcal{K}, i \in \{1, 2\}$.
24:    **end if**
25: **end while**

**Algorithm 2** Probing

1: Select the region of the branched variable $M_a^k$.
2: Divide $M_a^k$ into two sub-regions $M_{sub}$.
3: Compute basic LB for each sub-region.
4: Delete the sub-region if $\beta_{basic}(M_{sub}) > \alpha_l$.

**Algorithm 3** Cluster Assignment

1: **for** Sample $x_s \in X \cap M$, **do**
2:    **if** $x_s$ is not assigned in the parent node **then**
3:      **if** $d_{s,max}^{k^+} < d_{s,min}^{k^-}, \forall k^- \in \mathcal{K} \setminus k^+$ **then**
4:        Assign $x_s$ to $k^+$th cluster.
5:      **end if**
6:    **end if**
7: **end for**

**Algorithm 4** Feasibility Based Bound Tightening

1: **for** Cluster $k \in \mathcal{K}$ **do**
2:    Obtain the assigned set $\mathcal{S}_A^k$ according to Alg.3.
3:    Compute $RHS = \alpha_l - \beta(M)^-$.
4:    **for** Attribute $a \in \mathcal{A}$ **do**
5:      Obtain the tightened node, $\hat{M}_a^k = \{\mu_a^k | \hat{\underline{\mu}}_a^k \leq \mu_a^k \leq \hat{\bar{\mu}}_a^k\}$, by solving the inequation 13.
6:      Update $\hat{M}_a^k \leftarrow \hat{M}_a^k \cap M_a^k$
7:    **end for**
8: **end for**

**Algorithm 5** Lagrangian Based Lower Bounding

1: Compute basic LB, $\beta_{basic}(\hat{M})$.
2: **if** $|\beta_{basic}(\hat{M}) - \alpha_l| \geq \epsilon$ **then**
3:    Compute Lagrange Dual LB, $\beta_{LD}(\hat{M})$.
4:    $\beta(\hat{M}) = max\{\beta_{basic}(\hat{M}), \beta_{LD}(\hat{M})\}$.
5: **else**
6:    $\beta(\hat{M}) = \beta_{basic}(\hat{M})$.
7: **end if**

## 4.1 Lower Bounds

**Basic lower bound:** Besides the Lagrangian based lower bound, we also introduce a basic lower bound $\beta_{basic}$. Although it is not as tight as the Lagrangian based lower bound $\beta_{LD}$, its computational cost is significantly lower. Thus it is useful when the BB tree is deep with small medoid regions. For K-Medoids 1, there is an implicit constraint that all the samples share the same medoid set:

$$\mu_s^k = \mu_{s+1}^k, \forall s \in \{1, ..., S-1\}, k \in \mathcal{K}, \tag{10}$$

in which $\mu_s^k$ is the $k$th medoid for the $s$th sample. By removing the "medoids on samples" constraint $\mu \in X$ and Constraint 10, we have the basic lower bound $\beta_{basic}(M)$ with $\beta_{basic}(M) \leq z(M)$:

$$\beta_{basic}(M) = \min_{\mu_s \in M} \sum_{s \in S} \min_{k \in \mathcal{K}} ||x_s - \mu_s^k||_2^2 = \sum_{s \in S} \min_{k \in \mathcal{K}} \min_{\mu_s^k \in M^k} ||x_s - \mu_s^k||_2^2. \tag{11}$$

We define $d_{s,min}^k(M) := \min_{\mu^k \in M^k} ||x_s - \mu^k||_2^2$ and obviously $\beta_{basic}(M) = \sum_{s \in S} \min_{k \in \mathcal{K}} d_{s,min}^k(M)$. The analytic solution to $d_{s,min}^k(M)$ can be easily derived: $\mu_{s,a}^k = mid\{\underline{\mu}_a^k, x_{s,a}, \bar{\mu}_a^k\}, a \in \mathcal{A}$, in which $\underline{\mu}_a^k$ and $\bar{\mu}_a^k$ are the bounds of the $a$th attribute in $k$th medoid region.

**Lagrangian based lower bound:** as shown in Algorithm 5, by combining the basic and LD lower bound methods, we give the final form of lower bounds: $\beta(M) = max\{\beta_{basic}(M), \beta_{LD}(M)\}$. We note that because $\beta_{basic}(M) \leq z(M)$ and $\beta_{LD}(M) \leq z(M)$, we have $\beta(M) \leq z(M)$. Besides, according to the guarantee of maximum gap between the Lagrangian based lower bound and the global optimal value, we can also derive a guarantee of maximum gaps for $\beta(M)$:

**Proposition 2.** $\frac{z(M) - \beta(M)}{z_r - \beta(M)} < \frac{1}{e}$, where $z_r = \sum_{s \in \mathcal{S}} \max_{j \in \mathcal{S}} d_{s,j}$.

## 4.2 Upper Bounds

Upper bounds of the K-Medoids problem can be obtained by selecting a feasible set of medoids $\tilde{\mu} \in M \cap X$. Given a feasible solution, we can obtain the assignments of samples and object value through Equation 1, without solving any optimization problem explicitly. However, the selection of $\tilde{\mu}$ will critically influence the efficiency of convergence. Hence, in our implementation, we introduce two efficient upper bounding methods:

**Root node:** we use the K-Means-like method [4] as a backbone. Because the K-Means-like method is sensitive to the initial seeds and apt to the local minimum [5], we run this method several times with random seeds to get a candidate solution set. Then, we improve the quality of the upper bound by feeding the candidate solution set as initial guesses to Evolutionary Centers Algorithm [24].

**Child nodes:** we utilize the candidate solution obtained in the lower bounding method to update the upper bounds. Specifically, we first compute the corresponding assignments, then feed these assignments as the initial solution to the K-Means-like method. It should be noted that this K-Means-like method [4] gives a solution full-filling the "medoids on samples" constraint in K-Medoids.

# 5 Bound Tightening Techniques

Here, we introduce some bound tightening techniques tailored for the K-Medoids problem to reduce the search regions $M$ at each BB node while guaranteeing the optimal solution is not excluded. These bound tightening techniques are deployed before lower and upper bounds calculation.

## 5.1 Probing

Probing is a technique to reduce bound regions by exploiting the inequalities in a mixed integer programming problem [25]. This paper exploits the implicit inequality that the lower bound of a region containing the optimal solution is smaller than the current best upper bound. As shown in Algorithm 2, it tentatively restricts the bounds of $\mu$ to a subinterval and then computes the corresponding lower bound. If the lower bound is larger than the best upper bound, we can conclude that no optimal solution exists within the subinterval, and tighten the bound on the variable accordingly. The computation costs are trivial since we use the basic lower bounding method.

## 5.2 Feasibility Based Bound Tightening

This subsection introduces the Feasibility Based Bound Tightening (FBBT) technique. It is based on the observation that the assignment of some samples to clusters can be predetermined.

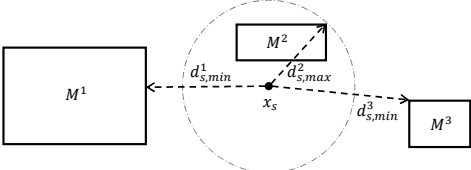

Figure 1: Cluster assignment with 3 clusters. In this example, we have $d^2_{s,max} < d^1_{s,min}$ and $d^2_{s,max} < d^3_{s,min}$. Therefore, we assign $x_s$ to the second cluster.

**Cluster assignment:** if we have $||x_s - \mu^{k^+}||^2_2 < ||x_s - \mu^{k^-}||^2_2, \forall k^- \in \mathcal{K} \setminus k^+$, then obviously sample $x_s$ is in the $k^+$th cluster. However, the value of $\mu$ here is not known before solving the overall problem. Recall that we have $d^k_{s,min}(M) := \min_{\mu^k \in M^k} ||x_s - \mu^k||^2_2$. Similarly, we also define $d^k_{s,max}(M) := \max_{\mu^k \in M^k} ||x_s - \mu^k||^2_2$ as the maximum distance between $x_s$ and $\mu_k$ with $\mu^k \in M^k$. At a BB node with partition $M$, if we have $d^{k^+}_{s,max}(M) < d^{k^-}_{s,min}(M), \forall k^- \in \mathcal{K} \setminus k^+$, then sample $x_s$ is guaranteed to be assigned to the $k^+$th cluster. Based on this observation, we can check whether the cluster of some samples can be determined in a BB node as shown in Algorithm 3. Recall that $d^k_{s,min}(M)$ is already computed as a byproduct of the basic lower bound in Section 3. Similarly, since the cluster regions for each node are rectangles, $d^k_{s,max}(M)$ can also be analytically computed. Besides, since child nodes are subsets of the parent node, once a sample is assigned to a cluster in

a parent node, it is also assigned to the same cluster in its child nodes. Figure 1 shows a simple illustration of this cluster assignment scheme.

**FBBT:** In a node with $M$, we denote the index sets of samples assigned to cluster $k$ as $\mathcal{S}_A^k$. FBBT infers the regions of $\mu^k$ from the following inequation:

$$\beta(M)^- + \sum_{s \in \mathcal{S}_A^k} ||x_s - \mu^k||_2^2 \leq \alpha_l, \forall k \in \mathcal{K} \tag{12}$$

where $\alpha_l$ is the current best upper bound, $\beta(M)^-$ represents the lower bound contributions of samples unassigned to cluster $k$ (either samples assigned to other clusters or samples whose assignments are undetermined). Similar to the calculation of the basic lower bounding method, we have $\beta(M)^- = \sum_{s \in \mathcal{S} \backslash \mathcal{S}_A^k} \min_{k \in \mathcal{K}} d_{s,min}^k(M)$.

As shown in Algorithm 4, FBBT is applied for each cluster. Specifically, for the $k$th medoid region, 1) obtain $\mathcal{S}_A^k$ according to the results of Algorithm 3; 2) compute the right-hand-side, $RHS = \alpha_l - \beta(M)^-$; 3) for each attribute $a \in \mathcal{A}$, obtain the tightened region, $\hat{M}_a^k = \{\mu_a^k | \hat{\underline{\mu}}_a^k \leq \mu_a^k \leq \hat{\overline{\mu}}_a^k\}$, by solving the following inequation:

$$\sum_{s \in \mathcal{S}_A^k} (x_{s,a} - \mu_a^k)^2 + \sum_{s \in \mathcal{S}_A^k} \sum_{i \in \mathcal{A} \backslash a} \min_{\mu_i^k \in M^k} (x_{s,i} - \mu_i^k)^2 \leq RHS, \quad \forall a \in \mathcal{A}, \forall k \in \mathcal{K}. \tag{13}$$

Since 13 is a simple quadratic inequation, we can analytically solve the problem to obtain the lower and upper bound of $\mu_a^k$ as $\hat{\underline{\mu}}_a^k$ and $\hat{\overline{\mu}}_a^k$. Finally, update $\hat{M}_a^k$ by $\hat{M}_a^k \leftarrow \hat{M}_a^k \cap M_a^k$.

### 5.3 Miscellaneous

**Symmetric-breaking:** we enforce a symmetric-breaking constraint on the first attribute, i.e., $\mu_1^k \leq \mu_1^{k+1}, \forall k = 1, ..., K - 1$. It is used to tighten the regions of $M$ at each BB node.

**Bound tightening according to the "medoids on samples" constraint:** since the K-Medoids problem requires medoids to be selected from samples in the dataset, we can further tighten the regions of $M$ by $\hat{M} = \{\mu | \hat{\underline{\mu}} \leq \mu \leq \hat{\overline{\mu}}\}$, $\hat{\underline{\mu}}_a^k = \min_{s \in M \cap X} x_{s,a}$, and $\hat{\overline{\mu}}_a^k = \max_{s \in M \cap X} x_{s,a}$.

### 5.4 Effects on Computation

With a reduced search space, these bound tightening techniques can significantly reduce the number of BB nodes to be explored. Moreover, they can also accelerate the calculation of bounds. For example, for the Lagrangian Problem 8, we need to calculate the contributions, $\rho_j$, of samples belonging to the overall feasible set of medoids, $\hat{S}^+ = \hat{S}^{1+} \cup \hat{S}^{2+} \cup ... \cup \hat{S}^{K+}$. A reduced search space $\hat{M}$ means a reduced overall feasible selection set, which further leads to fewer calculations and faster speeds.

## 6 Experiments and Discussions

To evaluate the performance, we use 25 datasets from the UCI Machine Learning Repository [26], one dataset called PR2392 from [27], one dataset called HEMI[28] and one synthetic dataset from [19]. The datasets from UCI Repository are in the clustering category with multivariate numerical attributes and range from 100 to 2,458,285 instances. We implement our branch and bound K-Medoids algorithm with Lagrangian based lower bound method (BB+LD) in Julia 1.6.1. We also implement four other methods to compare the performance, including the extensive form of K-Medoids problem solved by the state-of-art global optimizer CPLEX 20.1.0 [14] (CPLEX), the heuristic method in the root node (Heuristic), the pure Lagrangian relaxation method without branch and bound (LD), and the state-of-art BB method for K-Means in [19] adapting to K-Medoids by adding "medoids on samples" constraint (BB+Basic). We executed all the experiments on a high-performance computing cluster, of which each node contains 40 Intel cores at 2.4 GHz and 202 GB RAM. We present the results of cluster number $K = 3$ in the main body with additional results $K = 5$ and $K = 10$ in Appendix C. The complete code files can be found in https://github.com/YankaiGroup/global_kmedoids_clustering

The numerical performances are compared using three indicators, including upper bound value (UB), relative optimal gap, and the number of solved nodes. Here, UB is the best upper bound in all the iterations. The relative optimal gap is the relative difference between the best lower and upper bounds, which is calculated by $Gap(\%) := \frac{\alpha_l - \beta_l}{\min\{\alpha_l, \beta_l\}} \times 100\%$. This relative gap represents the worst gap of the found solution from the global optimal solution. Notably, this gap is a particular property of deterministic global optimization algorithms. Heuristic methods can not quantify the worst gap from the global optimal solution. The number of solved nodes represents the total number of BB nodes.

All the deterministic methods CPLEX, LD, BB+Basics, and BB+LD, share the same terminal criteria, including (1) $Gap \leq 0.1\%$, (2) the solving time reaches 4 hours, (3) the number of solved nodes reaches 5 million. Both LD and BB+LD use the same sub-gradient method called the Volume method [21]. Particularly, for BB+LD, the maximum iteration of Lagrangian multiplier updates in each node is 10. For LD, the maximum iteration is 10,000. In the heuristic method, we execute the K-Means-like method 20 times with a fixed seed set including 20 different seeds, then feed the solution set as initial guesses to ECA to get the best solutions.

Table 1: Serial numerical results ($K = 3$)

| DATA-SET | SAM-PLE | DIM-ENSION | METHOD | UB | NODES | GAP (%) | TIME (S) | DATA-SET | SAM-PLE | DIM-ENSION | METHOD | UB | NODES | GAP (%) | TIME (S) |
|---|---|---|---|---|---|---|---|---|---|---|---|---|---|---|---|
| IRIS | 150 | 4 | HEURISTIC | 84.63 | - | - | - | HEMI | 1955 | 7 | HEURISTIC | 9.91E+06 | - | - | - |
| | | | CPLEX | 83.91 | 1 | ≤0.10 | 41 | | | | CPLEX | 9.91E+06 | 1 | ≤0.10 | 2044 |
| | | | **LD** | **83.91** | **1** | **≤0.10** | **19** | | | | LD | 9.91E+06 | 1 | 12.45 | 3176 |
| | | | BB+Basic | 83.91 | 2.0E+5 | ≤0.10 | 124 | | | | BB+Basic | 9.91E+06 | 1.9E+6 | 4.32 | 4H |
| | | | BB+LD | 83.91 | 25 | ≤0.10 | 93 | | | | **BB+LD** | **9.91E+06** | **63** | **≤0.10** | **97** |
| SEEDS | 210 | 7 | HEURISTIC | 598.29 | - | - | - | PR2392 | 2392 | 2 | HEURISTIC | 2.13E+10 | - | - | - |
| | | | CPLEX | 598.29 | 1 | ≤0.10 | 60 | | | | CPLEX | 2.13E+10 | 1 | ≤0.10 | 5339 |
| | | | **LD** | **598.29** | **1** | **≤0.10** | **19** | | | | LD | 2.13E+10 | 1 | ≤0.10 | 35 |
| | | | BB+Basic | 598.29 | 2.4E+5 | ≤0.10 | 273 | | | | BB+Basic | 2.13E+10 | 3.0E+5 | 4.38 | 4H |
| | | | BB+LD | 598.29 | 9 | ≤0.10 | 84 | | | | **BB+LD** | **2.13E+10** | **37** | **≤0.10** | **123** |
| GLASS | 214 | 9 | HEURISTIC | 629.02 | - | - | - | TRR | 5456 | 24 | HEURISTIC | 1.96E+05 | - | - | - |
| | | | CPLEX | 629.02 | 1 | ≤0.10 | 60 | | | | CPLEX | - | - | - | - |
| | | | **LD** | **629.02** | **1** | **≤0.10** | **19** | | | | LD | 1.96E+05 | 1 | 0.41 | 4H |
| | | | BB+Basic | 629.02 | 2.2E+6 | ≤0.10 | 2871 | | | | BB+Basic | 1.96E+05 | 7.4E+4 | 1644.88 | 4H |
| | | | BB+LD | 629.02 | 32 | ≤0.10 | 107 | | | | **BB+LD** | **1.96E+05** | **553** | **≤0.10** | **325** |
| BM | 249 | 6 | HEURISTIC | 8.65E+05 | - | - | - | AC | 7195 | 22 | HEURISTIC | 2211.19 | - | - | - |
| | | | CPLEX | 8.63E+05 | 1 | ≤0.10 | 64 | | | | CPLEX | - | - | - | - |
| | | | **LD** | **8.63E+05** | **1** | **≤0.10** | **19** | | | | LD | 2199.12 | 1 | ≤0.10 | 287 |
| | | | BB+Basic | 8.63E+05 | 5.0E+6 | 25.74 | 7698 | | | | BB+Basic | 2200.04 | 1.3E+5 | 573.92 | 4H |
| | | | BB+LD | 8.63E+05 | 53 | ≤0.10 | 86 | | | | **BB+LD** | **2199.10** | **67** | **≤0.10** | **222** |
| UK | 258 | 5 | HEURISTIC | 50.77 | - | - | - | RDS_CNT | 10000 | 4 | HEURISTIC | 1.49E+07 | - | - | - |
| | | | **CPLEX** | **50.77** | **1** | **≤0.10** | **74** | | | | CPLEX | - | - | - | - |
| | | | LD | 50.77 | 1 | 0.96 | 176 | | | | LD | 1.49E+07 | 1 | 14.89 | 4H |
| | | | BB+Basic | 50.77 | 9.5E+5 | 3.93 | 4H | | | | BB+Basic | 1.49E+07 | 2.6E+5 | ≤0.10 | 5883 |
| | | | BB+LD | 50.77 | 119 | ≤0.10 | 89 | | | | **BB+LD** | **1.49E+07** | **31** | **≤0.10** | **203** |
| HF | 299 | 12 | HEURISTIC | 7.83E+11 | - | - | - | HTRU2 | 17898 | 8 | HEURISTIC | 8.21E+07 | - | - | - |
| | | | **CPLEX** | **7.83E+11** | **1** | **≤0.10** | **74** | | | | CPLEX | - | - | - | - |
| | | | LD | 7.83E+11 | 1 | 19.73 | 28 | | | | LD | 8.21E+07 | 1 | 22.97 | 4H |
| | | | BB+Basic | 7.83E+11 | 9.6E+4 | ≤0.10 | 173 | | | | BB+Basic | 8.21E+07 | 2.3E+5 | 25.48 | 4H |
| | | | BB+LD | 7.83E+11 | 99 | ≤0.10 | 107 | | | | **BB+LD** | **8.21E+07** | **465** | **≤0.10** | **1555** |
| WHO | 440 | 8 | HEURISTIC | 8.34E+10 | - | - | - | GT | 36733 | 11 | HEURISTIC | 1.95E+07 | - | - | - |
| | | | **CPLEX** | **8.33E+10** | **1** | **≤0.10** | **124** | | | | CPLEX | - | - | - | - |
| | | | LD | 8.33E+10 | 1 | 11.10 | 37 | | | | LD | 1.95E+07 | 1 | 1.43 | 4H |
| | | | BB+Basic | 8.33E+10 | 5.0E+6 | 2.42 | 11943 | | | | BB+Basic | 1.95E+07 | 4.3E+4 | 298.13 | 4H |
| | | | BB+LD | 8.33E+10 | 193 | ≤0.10 | 117 | | | | **BB+LD** | **1.95E+07** | **101** | **≤0.10** | **1936** |
| HCV | 572 | 12 | HEURISTIC | 2.75E+06 | - | - | - | RDS | 50000 | 3 | HEURISTIC | 486.75 | - | - | - |
| | | | **CPLEX** | **2.75E+06** | **1** | **≤0.10** | **121** | | | | CPLEX | - | - | - | - |
| | | | LD | 2.75E+06 | 1 | 0.34 | 536 | | | | LD | 486.75 | 1 | 4.34 | 4H |
| | | | BB+Basic | 2.75E+06 | 1.9E+6 | ≤0.10 | 5587 | | | | BB+Basic | 476.79 | 2.0E+5 | 35.72 | 4H |
| | | | BB+LD | 2.75E+06 | 4818 | ≤0.10 | 215 | | | | **BB+LD** | **476.79** | **23** | **≤0.10** | **811** |
| ABS | 740 | 21 | HEURISTIC | 2.62E+06 | - | - | - | KEGG | 53413 | 23 | HEURISTIC | 4.94E+08 | - | - | - |
| | | | CPLEX | 2.62E+06 | 23 | 0.23 | 285 | | | | CPLEX | - | - | - | - |
| | | | LD | 2.62E+06 | 1 | 0.86 | 726 | | | | LD | 4.94E+08 | 1 | 151.02 | 4H |
| | | | BB+Basic | 2.62E+06 | 2.4E+6 | 57.45 | 4H | | | | BB+Basic | 4.94E+08 | 1.8E+4 | 11.72 | 4H |
| | | | **BB+LD** | **2.62E+06** | **125** | **≤0.10** | **119** | | | | **BB+LD** | **4.94E+08** | **177** | **≤0.10** | **3901** |
| TR | 980 | 10 | HEURISTIC | 1136.93 | - | - | - | URBAN GB_10 | 100000 | 2 | HEURISTIC | 1.26E+05 | - | - | - |
| | | | CPLEX | 1134.45 | 3 | ≤0.10 | 855 | | | | CPLEX | - | - | - | - |
| | | | LD | 1136.93 | 1 | 0.80 | 130 | | | | LD | 1.26E+05 | 1 | 26.5704 | 4H |
| | | | BB+Basic | 1136.93 | 2.5E+6 | 209.23 | 4H | | | | BB+Basic | 1.15E+05 | 7.6E+4 | 17.86802 | 4H |
| | | | **BB+LD** | **1134.45** | **191** | **≤0.10** | **126** | | | | **BB+LD** | **1.15E+05** | **49** | **≤0.10** | **6834** |
| SGC | 1000 | 21 | HEURISTIC | 1.28E+09 | - | - | - | | | | | | | | |
| | | | CPLEX | 1.28E+09 | 1 | ≤0.10 | 347 | | | | | | | | |
| | | | LD | 1.28E+09 | 1 | 21.85 | 1248 | | | | | | | | |
| | | | BB+Basic | 1.28E+09 | 3.8E+5 | ≤0.10 | 2739 | | | | | | | | |
| | | | **BB+LD** | **1.28E+09** | **189** | **≤0.10** | **140** | | | | | | | | |

* OUT-OF-MEMORY IN THE ROOT NODE WITH NO SOLUTION.

## 6.1 Serial Results

Table 1 shows the serial numerical results for the datasets with less than 100,000 samples. In terms of best found upper bound (UB), the deterministic global optimization methods (CPLEX, BB+Basic, and BB+LD) can obtain better UB than the heuristic method in several datasets (e.g., IRIS, BM, RDS). While we try to obtain the best heuristic results by multiple initializations, there is still no guarantee for the heuristic method to reach the global optimum. Moreover, the heuristic method can not provide an optimal gap to estimate the solution quality. Notably, our method and CPLEX provide the same upper bounds and gaps for the datasets smaller than PR2392 (2392 samples).

As for the gaps and running times, CPLEX and LD can obtain a small gap ($\leq 0.1\%$) with less time when datasets are smaller than HCV (572 samples). However, when datasets are larger than HCV, our method, BB+LD, can reach the small gap with much less time than CPLEX and LD. Remarkably, our method can reach the small gap in all the datasets up to 100,000 samples within 4 hours. CPLEX fails to give a solution because of the out-of-memory error in the root node when datasets are larger than PR2392 (2392 samples). LD and BB+Basic methods may not reach the small gap within 4 hours or maximum iteration limits when datasets are larger than BM (249 samples). Recall that LD has no guarantee of reaching the global optimal solution because of the duality gap. All these results show that our method is extremely scalable for K-Medoids problems even under serial mode.

Notably, we followed the tuning guide in the CPLEX documentation to improve its performance. For example, we initiated the root node with the Heuristic solution and specified the node file and memory parameters. Moreover, we tried different MIP strategies, including start, pre-solve, and heuristic strategies. In Table 1, we present the best results obtained by CPLEX.

## 6.2 Parallel Results

For BB+LD, we executed parallel experiments utilizing MPI interfaces in the large and huge scale datasets with $K = 3$. The results of large-scale datasets ranging from 10,000 to 100,000 samples are presented in Appendix B. Table 2 shows the parallel numerical results for huge datasets with 100,000 to 2,458,285 samples. In this table, even for the dataset with 1,046,910 samples, BB+LD can still converge to the global optimal solution ($\leq 0.10$ gaps) within one hour, which is **400 times larger** than the size of problem that the state-of-art global solver CPLEX can address. For USC1990 with 2,458,285 samples and 68 features, BB+LD can obtain a 6.33% gap within 4 hours. According to Theorem 1, we can obtain a global $\epsilon$-optimal solution for USC1990 given more calculation time.

Table 2: Parallel results of huge scale datasets (BB+LD, $K = 3$)

| DATASET | SAMPLE | DIMENSION | CORES | UB | NODES | GAP(%) | TIME(S) |
|---------|--------|-----------|-------|-----|-------|--------|---------|
| RNG_AGR | 199,843 | 7 | 1600 | 8.23E+14 | 99 | $\leq 0.10$ | 341.0 |
| URBANGB | 360,177 | 2 | 1600 | 4.14E+05 | 57 | $\leq 0.10$ | 327.0 |
| SPNET3D | 434,874 | 3 | 1600 | 2.28E+07 | 115 | $\leq 0.10$ | 865.0 |
| RETAIL | 541,909 | 2 | 1600 | 6.80E+09 | 1 | $\leq 0.10$ | 80.0 |
| SYNTHETIC | 1,000,000 | 2 | 6000 | 9.44E+06 | 3 | $\leq 0.10$ | 171.0 |
| RETAIL-II | 1,046,910 | 2 | 6000 | 2.31E+10 | 214 | $\leq 0.10$ | 2515.0 |
| USC1990* | 2,458,285 | 68 | 3000 | 6.91E+08 | 25 | 6.33 | 4H |

$^*$ $d_{s,j}$ WAS COMPUTED ON-THE-FLY, NOT PRECOMPUTED AND STORED.

# 7 Complexity Analysis

## 7.1 Time Complexity

Denoting the number of samples as $S$, attributes as $A$, and clusters as $K$, our algorithm branches on the medoid regions to guarantee convergence, which contains $A \times K$ variables and is independent of the number of samples $S$. However, the number of BB nodes to converge is hard to predict since the efficiency of bounding methods depends on datasets. Also, the number of Lagrangian Dual (LD) iterations in each BB node is different because we set dynamic stopping criteria in case there is no update of LB in several continuous LD iterations. The time complexity of one LD iteration is $O(S^2)$ when $d_{s,j}$ are precomputed. If $d_{s,j}$ are not precomputed, the time complexity is $O(AS^2)$.

Table 3 compares computing time and nodes on synthetic datasets. In this table, the average runtime of one LD iteration remains almost the same as the dimension changes and increases almost

quadratically as the number of samples increases. As for the BB nodes, although the number of branching variables is $A \times K$, the actual number of explored BB nodes does not strictly follow the growth of dimensions. These results are consistent with the complexity analysis mentioned above.

Table 3: Comparison of time and nodes on synthetic datasets with varies dimensions and samples

| Samples | Dimension | 2 | 5 | 10 | 20 | 30 | 40 | 50 | 60 | 70 | 80 | 90 | 100 |
|---|---|---|---|---|---|---|---|---|---|---|---|---|---|
| 1,000 | Nodes* | 10 | 3 | 3 | 31 | 19 | 3 | 3 | 39 | 7 | 3 | 69 | 3 |
| | Time (ms)† | 4.28 | 4.28 | 4.06 | 3.57 | 3.62 | 4.02 | 4.11 | 3.78 | 4.32 | 3.8 | 3.21 | 3.79 |
| 10,000 | Nodes* | 3 | 17 | 7 | 17 | 17 | 15 | 41 | 13 | 7 | 3 | 201 | 17 |
| | Time (ms)† | 423 | 363 | 387 | 357 | 390 | 361 | 373 | 366 | 369 | 363 | 211 | 358 |
| 100,000 | Nodes* | 4 | 7 | 27 | 31 | 3 | 15 | 27 | 13 | 25 | 7 | 25 | 21 |
| | Time (s)† | 40.6 | 39.2 | 34.3 | 33.4 | 35.2 | 28.7 | 35.6 | 37.1 | 34.6 | 36.2 | 35.2 | 34.3 |

\* The number of solved nodes in the BB procedure until convergence.
† Time consumed in each Lagrangian Dual iteration.

## 7.2 Space Complexity of Distance

When distances $d_{s,j}$ are precomputed and stored, the space complexity of $d_{s,j}$ matrix is $O(S^2)$. In our implementation, the datasets with no more than 100,000 samples were computed on one compute node with 40 cores and 202GB RAM. For more than 100,000 samples, we executed the experiments on multiple compute nodes. Each core precomputes and stores part of the $d_{s,j}$ matrix, as described in Section 3. For example, we executed the one-million dataset RETAIL-II on 150 nodes with 6,000 cores. Each core only needs to store 1.25GB of $d_{s,j}$ matrix. For more than 1,000,000 samples, we calculated $d_{s,j}$ on the fly, without precomputing and storing. In this case, the time complexity of one LD iteration increases from $O(S^2)$ to $O(AS^2)$. Hence, we expect an acceptable slowdown when the dimension of datasets is small. For example, in Table 2, the dataset USC1990 with two-million samples and 68 features was executed without precomputing $d_{s,j}$, and the result is acceptable. Table 4 compares the computing time of precomputed distance and on-the-fly computing. We can conclude that although calculating $d_{s,j}$ on the fly is slower than precomputing $d_{s,j}$, the slowdown is acceptable.

Table 4: Comparison of precomputed distance and on the fly computing

| Dataset | Dimension | Total Run Time (s) | | Time per LD iteration (ms) | |
|---|---|---|---|---|---|
| | | On-the-fly | Precomputed | On-the-fly | Precomputed |
| ABS | 21 | 34 | 12 | 1.42 | 0.70 |
| HEMI | 7 | 17 | 13 | 9.91 | 2.02 |
| RDS_CNT | 4 | 47 | 30 | 175 | 37.80 |
| TR | 10 | 58 | 15 | 2.54 | 0.73 |
| TRR | 24 | 365 | 74 | 110 | 10.80 |

# 8 Conclusion

We presented a scalable global optimization algorithm of the K-Medoids problem by applying a tailored reduced-space spatial branch and bound scheme. This algorithm includes a Lagrangian based lower bounding method and a basic lower bounding method. Bound tightening techniques are also proposed to accelerate the solution process. We demonstrate our algorithm's scalability by extensive numerical experiments and prove the convergence by theoretical analysis. Besides, one interesting finding is that our algorithm can reduce the objective values for some datasets while remaining the same as the heuristic method for most datasets. We also compared other clustering evaluation metrics in Appendix D, such as Normalized Mutual Information (NMI) and Adjusted Rand Index (ARI). A similar conclusion with objective values holds for these metrics. However, it should be noted that only with the global optimal solution obtained in our work can we give a fair comparison of the heuristic methods and, in turn, contribute to developing a more efficient heuristic method.

## Acknowledgments and Disclosure of Funding

The authors acknowledge funding from the discovery program of the Natural Science and Engineering Research Council of Canada under grant RGPIN-2019-05499 and the computing resources provided by SciNet (www.scinethpc.ca) and Digital Research Alliance of Canada (www.alliancecan.ca). Jiayang Ren acknowledges the financial support from the China Scholarship Council.

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
