# Appendix

## A  Convergence Analysis

As stated in Algorithm 1, the branch and bound scheme is a rooted tree, where the search space of the level 0 root node is $\overset{\circ}{M}$. We denote $M_{l_q}$ as a sub-node at iteration $l_q$ and level $q$. Its child node is denoted as $M_{l_{q+1}}$ satisfying $M_{l_{q+1}} \subset M_{l_q}$. A decreasing sequence from the root node $\overset{\circ}{M}$ to the node $M_{l_q}$ is denoted as $\{M_{l_q}\}$. It is obvious that the sequences $\{\alpha_l\}$ and $\{\beta_l\}$ are monotonically non-increasing and non-decreasing correspondingly. In the following convergence analysis, we adapt the fundamental results from [29] to our work. Different from the BB procedure in [19], which is infinite, our algorithm has a finite BB procedure. Hence, we prove the convergence in a finite manner.

**Definition 3.** (Definition IV.3 [29]) A bounding operation is called **finitely consistent** if, at every step, any unfathomed partition element can be further refined, and if any decreasing sequence $\{M_{l_q}\}$ successively refined partition elements is finite.

**Lemma 4.** *The bounding operation in Algorithm 1 is finitely consistent.*

*Proof.* We first prove any unfathomed partition element $M_{l_q}$ can be further refined. Any unfathomed $M_{l_q}$ satisfies $\exists |M_{l_q} \cap X| > 1, k \in \mathcal{K}$ and $\alpha_l > \beta(M_{l_q}) + \epsilon, \epsilon > 0$. It is obvious that there exists at least one partition to be further refined.

We then prove any decreasing sequence $\{M_{l_q}\}$ successively refined partition elements is finite. Assuming by contradiction that there exists a sequence $\{M_{l_q}\}$ that is infinite. In our algorithm, since we branch on the first-stage variable $\mu^k$ corresponding to the diameter of $M$, this subdivision is exhaustive, we have $\lim_{q \to \infty} \delta(M_{l_q}) = 0$ and $\{M_{l_q}\}$ converge to one point $\bar{\mu}$. If $\bar{\mu} \in X$, there is a ball around $\bar{\mu}$, denoted as $B_r(\bar{\mu}) = \{\mu \mid ||\mu - \bar{\mu}|| \leq r\}$, satisfying $|B_r(\bar{\mu}) \cap X| = 1$. There exists a $q_0$ that $M_{l_q} \subset B_r(\bar{\mu}), \forall q \leq q_0$. At this $q_0$ iteration, $M_{l_{q_0}}$ will not be branched anymore. Because $X$ is finite, we have the sequence $\{M_{l_q}\}$ is finite in this case. If $\bar{\mu} \not\subset X$, there is a ball around $\bar{\mu}$, denoted as $B_r(\bar{\mu}) = \{\mu \mid ||\mu - \bar{\mu}|| \leq r\}$, satisfying $|B_r(\bar{\mu}) \cap X| = 0$. There exists a $q_0$ that $M_{l_q} \subset B_r(\bar{\mu}), \forall q \leq q_0$. At this $q_0$ iteration, $M_{l_{q_0}}$ will be deleted. Consequently, in this case, the sequence $\{M_{l_q}\}$ is finite. Hence, it is impossible to exists a sequence $\{M_{l_q}\}$ that is infinite. $\square$

**Theorem 5.** *(Theorem IV.1 [29]) In a BB procedure, suppose that the bounding operation is finitely consistent. Then the procedure terminates after finitely many steps.*

**Lemma 6.** *Algorithm 1 terminates after finitely many steps*

*Proof.* From Lemma 4, we have the bounding operation in Algorithm 1 is finitely consistent. From Theorem 5, Algorithm 1 terminates after finitely many steps $\square$

Finally, we prove that the BB scheme is convergent, as shown in Theorem 1:

**Theorem 1.** *Algorithm 1 is convergent to the global optimal solution after a finite step $L$, with $\beta_L = z = \alpha_L$, by only branching on the space of $\mu$.*

*Proof.* From Lemma 6, we have Algorithm 1 terminates after finite steps. The algorithm terminates with two situations. The first situations is $|\beta_l - \alpha_l| \leq \epsilon$. When $\epsilon = 0$, we have $\beta_l = z = \alpha_l$.

The second situations is $\mathbb{M} = \emptyset$. A node $M$ is deleted from $\mathbb{M}$ and not further partitioned either because $\beta(M) > \alpha_l$ or $|M^k \cap X| = 1, \forall k \in \mathcal{K}$. The first case obviously does not contain the global optimal solution $\mu^*$. Therefore, the node $M'$ containing $\mu^*$, is not further partitioned because $|M'^k \cap X| = 1, \forall k \in \mathcal{K}$. After bound tightening according to the "medoids on samples" constraint, the tightened node $M' = \{\mu^*\}$. Obviously for this node, we have $\beta_l = \beta(M') = z = \alpha(M') = \alpha_l$. Consequently, we have proved Theorem 1.

$\square$

# B  Parallel results of large scale datasets

Table 5 shows the parallel numerical results for large scale datasets with 10,000 to 2,458,285 samples. For the large-scale datasets ranging from 10,000 to 100,000 samples, we obtain an average of 13.80x speedup in time with 40 cores. Here, speedup ratio is defined as $\frac{\bar{T}_1}{\bar{T}_c}$, in which $\bar{T}_c$ is the overall calculation time with $c$ cores. The average efficiency is low since the time ratio of the parallel part is relatively low in the small datasets. Consequently, we can expect a higher speedup in the big datasets. For example, we obtain a 25.89x $Speedup$ for URBANGB_10 with 100,000 samples compared to 7.57x for HTRU2 with 17,898 samples.

Table 5: Parallel results of large scale datasets (BB+LD, $k = 3$)

| DATASET | SAMPLE | DIMENSION | CORES | UB | NODES | GAP(%) | TIME(S) |
|---|---|---|---|---|---|---|---|
| HTRU2 | 17,898 | 8 | 40 | 8.21E+07 | 465 | ≤0.10 | 205 |
| GT | 36,733 | 11 | 40 | 1.95E+07 | 101 | ≤0.10 | 161 |
| RDS | 50,000 | 3 | 40 | 476.792 | 23 | ≤0.10 | 71 |
| KEGG | 53,413 | 23 | 40 | 4.94E+08 | 177 | ≤0.10 | 322 |
| URBANGB_10 | 100,000 | 2 | 40 | 1.15E+05 | 49 | ≤0.10 | 264 |

# C  Additional results of K-Medoids problems with $K = 5$ and $K = 10$

In this section, we perform several additional experiments with more clusters on the datasets ranging from 100 to 100,000 samples. All these experiments use the same setup in Section 6. Table 6 and 7 are the results of the BB+LD method with cluster number $K = 5$ and $K = 10$ respectively. When the cluster number is bigger, our BB+LD are more likely to obtain an upper bound smaller than the Heuristic UB. However, the difference in UB between BB+LD and Heuristic becomes smaller. Moreover, the search space of medoids increases as the cluster number, making it harder to obtain the global optimum. Nevertheless, our BB+LD method can still obtain a relative gap smaller than 0.1% within 4 hours for most datasets when $K = 5$ and a reasonable relative gap within 4 hours when $K = 10$.

Table 6: Additional results of K-Medoids problems with BB+LD and $K = 5$

| DATASET | SAM-PLE | DIM-ENSION | HEURISTIC UB | SERIAL RESULTS (CORE=1) | | | | PARALLEL RESULTS (CORE=40) | | | |
|---|---|---|---|---|---|---|---|---|---|---|---|
| | | | | UB | NODES | GAP (%) | TIME (S) | UB | NODES | GAP (%) | TIME (S) |
| IRIS | 150 | 4 | 5.1000E+01 | 5.0920E+01 | 88 | ≤0.10 | 355 | - | - | - | - |
| SEEDS | 210 | 7 | 4.0372E+02 | 4.0121E+02 | 43 | ≤0.10 | 376 | - | - | - | - |
| GLASS | 214 | 9 | 4.3789E+02 | 4.3773E+02 | 6,846 | ≤0.10 | 592 | - | - | - | - |
| BM | 249 | 6 | 6.0249E+05 | 6.0249E+05 | 281 | ≤0.10 | 389 | - | - | - | - |
| UK | 258 | 5 | 4.0166E+01 | 4.0166E+01 | 1,869 | ≤0.10 | 457 | - | - | - | - |
| HF | 299 | 12 | 3.0998E+11 | 3.0998E+11 | 43,827 | ≤0.10 | 1723 | - | - | - | - |
| WHO | 440 | 8 | 5.5914E+10 | 5.5914E+10 | 32,832 | ≤0.10 | 1840 | - | - | - | - |
| HCV | 572 | 12 | 1.9716E+06 | 1.9716E+06 | 1,011,564 | ≤0.10 | 12768 | - | - | - | - |
| ABS | 740 | 21 | 1.7476E+06 | 1.7472E+06 | 329 | ≤0.10 | 410 | - | - | - | - |
| TR | 980 | 10 | 9.6555E+02 | 9.5339E+02 | 3,975 | ≤0.10 | 824 | - | - | - | - |
| SGC | 1,000 | 21 | 4.6969E+08 | 4.6919E+08 | 23,827 | ≤0.10 | 3290 | 4.6919E+08 | 23,827 | ≤0.10 | 880 |
| HEMI | 1,955 | 7 | 5.3864E+06 | 5.3811E+06 | 1,782 | ≤0.10 | 930 | 5.3811E+06 | 1,782 | ≤0.10 | 66 |
| PR2392 | 2,392 | 2 | 1.1620E+10 | 1.1619E+10 | 405 | ≤0.10 | 625 | 1.1619E+10 | 405 | ≤0.10 | 41 |
| TRR | 5,456 | 24 | 1.6991E+05 | 1.6870E+05 | 3,029 | ≤0.10 | 3094 | 1.6870E+05 | 3,029 | ≤0.10 | 499 |
| AC | 7,195 | 22 | 1.6377E+03 | 1.6361E+03 | 1,473 | ≤0.10 | 3552 | 1.6361E+03 | 1,473 | ≤0.10 | 356 |
| RDS_CNT | 10,000 | 4 | 5.3725E+06 | 5.3725E+06 | 4,051 | ≤0.10 | 7171 | 5.3725E+06 | 4,051 | ≤0.10 | 450 |
| HTRU2 | 17,898 | 8 | 4.2154E+07 | 4.2154E+07 | 3,453 | 10.85 | 4H | 4.2154E+07 | 33,682 | 2.34 | 4H |
| GT | 36,733 | 11 | 1.3358E+07 | 1.3351E+07 | 669 | 0.97 | 4H | 1.3351E+07 | 4,095 | ≤0.10 | 4776 |
| RDS | 50,000 | 3 | 2.8452E+02 | 2.8265E+02 | 499 | 0.94 | 4H | 2.8265E+02 | 893 | ≤0.10 | 1338 |
| KEGG | 53,413 | 23 | 1.9201E+08 | 1.9201E+08 | 503 | 24.54 | 4H | 1.9200E+08 | 8,667 | 1.75 | 4H |
| URBANGB_10 | 100,000 | 2 | 5.6232E+04 | 5.6232E+04 | 104 | 3.63 | 4H | 5.6232E+04 | 543 | ≤0.10 | 2266 |

Table 7: Additional results of K-Medoids problems with BB+LD and $K = 10$

| DATASET | SAM-PLE | DIM-ENSION | HEURISTIC UB | SERIAL RESULTS (CORE=1) | | | | PARALLEL RESULTS (CORE=40) | | | |
|---|---|---|---|---|---|---|---|---|---|---|---|
| | | | | UB | NODES | GAP (%) | TIME (S) | UB | NODES | GAP (%) | TIME (S) |
| IRIS | 150 | 4 | 3.0380E+01 | 2.9790E+01 | 6,219 | ≤0.10 | 735 | - | - | - | - |
| SEEDS | 210 | 7 | 2.1849E+02 | 2.1452E+02 | 919 | ≤0.10 | 448 | - | - | - | - |
| GLASS | 214 | 9 | 2.5325E+02 | 2.5186E+02 | 31,983 | ≤0.10 | 2566 | - | - | - | - |
| BM | 249 | 6 | 3.8181E+05 | 3.7597E+05 | 10,965 | ≤0.10 | 1204 | - | - | - | - |
| UK | 258 | 5 | 2.9785E+01 | 2.9280E+01 | 176,103 | 1.70 | 4H | - | - | - | - |
| HF | 299 | 12 | 6.9604E+10 | 6.9604E+10 | 132,742 | 21.96 | 4H | - | - | - | - |
| WHO | 440 | 8 | 3.4614E+10 | 3.4020E+10 | 107,430 | 11.13 | 4H | - | - | - | - |
| HCV | 572 | 12 | 1.1592E+06 | 1.1315E+06 | 77,009 | 7.20 | 4H | - | - | - | - |
| ABS | 740 | 21 | 1.1083E+06 | 1.0786E+06 | 56,090 | 0.19 | 4H | - | - | - | - |
| TR | 980 | 10 | 7.7370E+02 | 7.7247E+02 | 41,752 | 3.17 | 4H | - | - | - | - |
| SGC | 1,000 | 21 | 1.1742E+08 | 1.1742E+08 | 33,388 | 20.70 | 4H | 1.1742E+08 | 254,583 | 16.33 | 4H |
| HEMI | 1,955 | 7 | 2.7421E+06 | 2.7421E+06 | 16,711 | 9.69 | 4H | 2.7068E+06 | 351,441 | 0.15 | 4H |
| PR2392 | 2,392 | 2 | 5.3578E+09 | 5.3578E+09 | 8,706 | 4.97 | 4H | 5.3578E+09 | 195,660 | 0.52 | 4H |
| TRR | 5,456 | 24 | 1.3933E+05 | 1.3796E+05 | 2,487 | 0.18 | 4H | 1.3796E+05 | 22,527 | ≤0.10 | 4H |
| AC | 7,195 | 22 | 1.1817E+03 | 1.1817E+03 | 1,341 | 3.16 | 4H | 1.1637E+03 | 32,530 | 0.63 | 4H |
| RDS_CNT | 10,000 | 4 | 1.6119E+06 | 1.6119E+06 | 1,020 | 26.13 | 4H | 1.6119E+06 | 33,615 | 13.15 | 4H |
| HTRU2 | 17,898 | 8 | 1.8273E+07 | 1.8273E+07 | 350 | 24.39 | 4H | 1.8273E+07 | 16,083 | 16.86 | 4H |
| GT | 36,733 | 11 | 8.9909E+06 | 8.9909E+06 | 65 | 4.89 | 4H | 8.9909E+06 | 5,380 | 1.13 | 4H |
| RDS | 50,000 | 3 | 1.3273E+02 | 1.3273E+02 | 40 | 7.89 | 4H | 1.3273E+02 | 4,756 | 3.79 | 4H |
| KEGG | 53,413 | 23 | 6.1564E+07 | 6.1564E+07 | 51 | 67.53 | 4H | 6.1564E+07 | 919 | 30.11 | 4H |
| URBANGB_10 | 100,000 | 2 | 2.5123E+04 | 2.5123E+04 | 14 | 23.54 | 4H | 2.5123E+04 | 1,427 | 9.04 | 4H |

# D   Comparison of heuristic methods for K-Medoids problems

To illustrate the effectiveness of our upper bound method (Heuristic UB), we compares its performance with several popular heuristic methods in the literature, including Kmeans, Kmeans++, and PAM. Here, we run all the methods several times with random seeds and select the best result. The centers of K-Means and K-Means++ are projected to the nearest samples to fulfill the "Medoids on Samples" constraint in the KMedoids problem. This table shows that the Heuristic UB can always obtain the same or better objective value than Kmeans, Kmeans++, and PAM.

Table 8: Additional results of heuristic methods for K-Medoids problems

| DATASET | KMEANS | KMEANS++ | PAM | HEURISTIC | BB+LD |
|---|---|---|---|---|---|
| IRIS | 84.63 | 84.63 | 90.99 | 84.63 | **83.91** |
| SEEDS | 598.29 | 598.29 | 608.72 | 598.29 | 598.29 |
| GLASS | 629.02 | 629.02 | 652.15 | 629.02 | 629.02 |
| BM | 8.65E+05 | 8.65E+05 | 9.17E+05 | 8.65E+05 | **8.63E+05** |
| UK | 50.77 | 51.19 | 51.06 | 50.77 | 50.77 |
| HF | 7.83E+11 | 7.83E+11 | 7.83E+11 | 7.83E+11 | 7.83E+11 |
| WHO | 8.34E+10 | 8.34E+10 | 8.44E+10 | 8.34E+10 | **8.33E+10** |
| HCV | 2.85E+06 | 2.82E+06 | 2.76E+06 | 2.75E+06 | 2.75E+06 |
| ABS | 2.62E+06 | 2.62E+06 | 2.66E+06 | 2.62E+06 | 2.62E+06 |
| TR | 1.14E+03 | 1.14E+03 | 1.16E+03 | 1.14E+03 | **1.13E+03** |
| SGC | 1.28E+09 | 1.28E+09 | 1.49E+09 | 1.28E+09 | 1.28E+09 |
| HEMI | 9.92E+06 | 9.91E+06 | 1.18E+07 | 9.91E+06 | 9.91E+06 |
| PR2392 | 2.13E+10 | 2.13E+10 | 2.53E+10 | 2.13E+10 | 2.13E+10 |
| TRR | 1.97E+05 | 1.96E+05 | 1.97E+05 | 1.96E+05 | 1.96E+05 |
| AC | 2.21E+03 | 2.21E+03 | 2.34E+03 | 2.21E+03 | **2.20E+03** |
| RDS_CNT | 1.49E+07 | 1.49E+07 | 1.50E+07 | 1.49E+07 | 1.49E+07 |
| HTRU2 | 8.21E+07 | 8.21E+07 | 8.61E+07 | 8.21E+07 | 8.21E+07 |
| GT | 1.95E+07 | 1.95E+07 | 1.96E+07 | 1.95E+07 | 1.95E+07 |
| RDS | 476.88 | 476.88 | 486.75 | 476.88 | **476.79** |
| KEGG | 4.94E+08 | 4.94E+08 | 4.95E+08 | 4.94E+08 | 4.94E+08 |
| URBANGB_10 | 1.15E+05 | 1.15E+05 | 1.26E+05 | 1.15E+05 | 1.15E+05 |
| RNG_AGR | 8.23E+14 | 8.23E+14 | -* | 8.23E+14 | 8.23E+14 |
| URBANGB | 4.14E+05 | 4.14E+05 | -* | 4.14E+05 | 4.14E+05 |
| SPNET3D | 2.28E+07 | 2.28E+07 | -* | 2.28E+07 | 2.28E+07 |
| RETAIL | 6.80E+09 | 6.80E+09 | -* | 6.80E+09 | 6.80E+09 |
| SYNTHETIC | 9.44E+06 | 9.44E+06 | -* | 9.44E+06 | 9.44E+06 |
| RETAIL-II | 2.90E+10 | 2.31E+10 | -* | 2.31E+10 | 2.31E+10 |
| USC1990 | 6.91E+08 | 6.91E+08 | -* | 6.91E+08 | 6.91E+08 |

* OUT-OF-MEMORY ON ONE COMPUTER NODE

We also compared other clustering evaluation metrics, such as Normalized Mutual Information (NMI) and Adjusted Rand Index (ARI). Since these clustering evaluation metrics need to compare the predicted cluster with ground truth, we can only compare the datasets with ground truth, such as IRIS, HCV, HF, HRTU2 and UK. Comparing Table 8 and Table 9, one interesting finding is that our algorithm can reduce the objective values for some datasets while remaining the same as the heuristic method for most datasets. A similar conclusion with objective values also holds for ARI and NMI. However, it should be noted that only by comparing the heuristic solution with the global optimal solution as we did in the table, we can confidently claim that the heuristic method can do a fairly nice job in finding near-optimal solutions.

Table 9: ARI and NMI of heuristic methods for K-Medoids problems

| DATASET | SAMPLE | DIMENSION | CLUSTER NUMBER (GROUND TRUTH) | METHOD | UB (OBJECT VALUE) | ARI | NMI |
|---|---|---|---|---|---|---|---|
| IRIS | 150 | 4 | 3 | KMEANS | 8.4680E+01 | 0.7302 | 0.7582 |
| | | | | KMEANS++ | 8.4680E+01 | 0.7302 | 0.7582 |
| | | | | PAM | 9.1040E+01 | 0.7060 | 0.7561 |
| | | | | HEURISTIC | 8.4680E+01 | 0.7302 | 0.7582 |
| | | | | BB+LD | **8.3960E+01** | **0.7455** | **0.7980** |
| HCV | 572 | 12 | 4 | KMEANS | 2.2873E+06 | 0.4417 | 0.2454 |
| | | | | KMEANS++ | 2.2873E+06 | 0.4417 | 0.2454 |
| | | | | PAM | 2.3168E+06 | **0.5941** | **0.3363** |
| | | | | HEURISTIC | 2.2873E+06 | 0.4417 | 0.2454 |
| | | | | BB+LD | 2.2873E+06 | 0.4417 | 0.2454 |
| HF | 299 | 12 | 2 | KMEANS | 1.3512E+12 | 0.0175 | 0.0025 |
| | | | | KMEANS++ | 1.3512E+12 | 0.0175 | 0.0025 |
| | | | | PAM | 1.3917E+12 | 0.0122 | 0.0013 |
| | | | | HEURISTIC | 1.3512E+12 | 0.0175 | 0.0025 |
| | | | | BB+LD | 1.3512E+12 | 0.0175 | 0.0025 |
| HRTU2 | 17898 | 8 | 2 | KMEANS | 1.2536E+08 | -0.0780 | 0.0267 |
| | | | | KMEANS++ | 1.2536E+08 | -0.0780 | 0.0267 |
| | | | | PAM | 1.2800E+08 | -0.0738 | 0.0221 |
| | | | | HEURISTIC | 1.2536E+08 | -0.0780 | 0.0267 |
| | | | | BB+LD | **1.2535E+08** | **-0.0779** | **0.0271** |
| UK | 258 | 5 | 4 | KMEANS | 4.5044E+01 | 0.2378 | 0.3253 |
| | | | | KMEANS++ | 4.5044E+01 | 0.2378 | 0.3253 |
| | | | | PAM | 4.5420E+01 | 0.1539 | 0.2210 |
| | | | | HEURISTIC | 4.5040E+01 | 0.2378 | 0.3253 |
| | | | | BB+LD | 4.5040E+01 | 0.2378 | 0.3253 |