# OpenReview forum: "Global Optimal K-Medoids Clustering of One Million Samples"
_NeurIPS.cc/2022/Conference — NeurIPS 2022 Accept_

### Official Review · Reviewer_FDa1 · 2022-07-11

**Rating:** 5
**Confidence:** 3
**Soundness:** 3 good
**Presentation:** 2 fair
**Contribution:** 2 fair

**Summary:**

This paper uses a branch-and-bound (BB) method to improve Cornuejols work on Lagrangian relaxation for the K-Medoids clustering problem. Through bounding the regions of the medoids, search space is significantly reduced, leading to notable computation speed improvements. A convergence proof is also provided. To obtain the lower and upper bound, a couple of tricks are used: 1. Other than the Lagrangian based lower bound, the author proposes a simple basic lower bound, to further speed up the process. 2. The author proposes heuristic methods to obtain the upper bound; 3. Different bound tightening methods (probing, FBBT) are applied before updating the search space. The proposed approach can run on millions of data samples within reasonable time.


**Questions:**

It would be better to introduce more on BB method in this setting, e.g. what are the BB tree and node? How are they constructed?

**Strengths And Weaknesses:**

Strengths:
1. The combination of brand-and-bounding with Lagrangian relaxation, on the K-Medoids problem is relatively novel. This method makes the Lagrangian based K-Medoids approach much more applicable in real-world datasets that have more than thousands of samples.
2. The proposed approach is trivially parallelizable, which makes the method suitable for large scale problems.
3. Theoretical convergence proof is provided.

Weaknesses:
The biggest weakness of the paper is regarding the practical value of the proposed method. It might interest the optimization community due to its convergence guarantee and various bound-shrinking tricks, but as a clustering method, the proposed approach is significantly more complicated than the counterparts like K-Means:
1. Though vanilla K-Means does not converge to global optimal, simple modifications like K-Means++ could alleviate the problem and often performs well in practice. The paper lacks a comparison (in terms of speed and quality) with such a simple baseline (K-Means or K-Means style methods).
2. Other than the centroid-based approaches, there are many other kinds of clustering, e.g. density-based, hierarchical clustering, etc. They often don't need the ball-shape assumption and work better in real world datasets. The paper lacks comparison with these different approaches, especially the state-of-the-art ones like [Sarfraz, et al, Efficient Parameter-free Clustering Using First Neighbor Relations].
3.  In experiments, the performance are evaluated using the achieved optimal gap and number of solved nodes. Standard clustering metrics include ARI and NMI, which provide a sense of clustering quality. The gap is useful in theoretical settings but not very meaningful in applications.

------

Score update from borderline reject to borderline accept, after reviewing the authors' responses.

---

> ### Author Response · Authors · 2022-08-02
> **Response to Reviewer FDa1**
>
> We sincerely thank the reviewer for their time and thoughtful comments. We provide several clarifications addressing the reviewer's main concerns below.
>
> * **Q1 - The biggest weakness of the paper is regarding the practical value of the proposed method. It might interest the optimization community due to its convergence guarantee and various bound-shrinking tricks, but as a clustering method, the proposed approach is significantly more complicated than the counterparts like K-Means.**
>
>     Although we agree with the reviewer that K-Means and K-Medoids are similar, the main difference is that K-Medoids enforces the "medoids on samples" constraint. This constraint requires k-medoids to choose actual data points as centers (medoids), and thereby leads to greater interpretability of the cluster centers than K-Means, where the centers can be arbitrary points. This constraint is essential for many applications, such as facility locations and anomaly detection. For facility locations, we expect the selected location to be existing locations (one of the input data points) rather than any random location where it is impossible to build the facility. For anomaly detection of industry processes, we expect the center of a normal operation condition to be the actual existing operation condition rather than any nonexistent condition.
>
>     Moreover, our algorithm provides a global optimal solution with an optimal gap. On the one hand, the optimal gap can help users understand the current solution's quality and decide whether to invest more time and effort to get a better solution. On the other hand, the global optimal solution from our work can give the community a baseline to evaluate the performance of a heuristic method and, in turn, contributes to developing a more efficient heuristic method.
>
> * **Q2 - Though vanilla K-Means does not converge to global optimal, simple modifications like K-Means++ could alleviate the problem and often performs well in practice. The paper lacks a comparison (in terms of speed and quality) with such a simple baseline (K-Means or K-Means style methods).**
>
>     Thank you for suggesting alternative methods of comparison. In our original submission, we have chosen the best baseline (we call it HEURISTIC) to compare with. This Heuristic uses the K-Means-like method proposed by Park and Jun (2009) as a backbone and then improves the quality of the upper bound using the Evolutionary Centers Algorithm.
>
>     We agree with the reviewer that it would have been interesting to also compare with K-Means and K-Means++. Therefore, We performed additional numerical experiments to compare the HEURISTIC, K-Means, K-Means++, and PAM (another popular method in the literature). Here, we run all the methods many times with random seeds and select the best result. The centers of K-Means and K-Means++ are projected to the nearest samples to fulfill the "Medoids on Samples" constraint. The following table shows that the HEURISTIC used in our paper can always obtain the same or better objective value than Kmeans, Kmeans++, and PAM. Moreover, BB+LD (our global optimal algorithm) can further improve the objective value for several datasets.
>
> |dataset|Kmeans|Kmeans++|PAM|HEURISTIC|BB+LD|
> |:---:|:---:|:---:|:---:|:---:|:---:|
> |iris|84.63|84.63|90.99|84.63|**83.91**|
> |seeds|598.29|598.29|608.72|598.29|598.29|
> |glass|629.02|629.02|652.15|629.02|629.02|
> |BM|8.65E+05|8.65E+05|9.17E+05|8.65E+05|**8.63E+05**|
> |UK|50.77|51.19|51.06|50.77|50.77|
> |HF|7.83E+11|7.83E+11|7.83E+11|7.83E+11|7.83E+11|
> |Who|8.34E+10|8.34E+10|8.44E+10|8.34E+10|**8.33E+10**|
> |HCV|2.85E+06|2.82E+06|2.76E+06|2.75E+06|2.75E+06|
> |Abs|2.62E+06|2.62E+06|2.66E+06|2.62E+06|2.62E+06|
> |TR|1.14E+03|1.14E+03|1.16E+03|1.14E+03|**1.13E+03**|
> |SGC|1.28E+09|1.28E+09|1.49E+09|1.28E+09|1.28E+09|
> |hemi|9.92E+06|9.91E+06|1.18E+07|9.91E+06|9.91E+06|
> |pr2392|2.13E+10|2.13E+10|2.53E+10|2.13E+10|2.13E+10|
> |TRR|1.97E+05|1.96E+05|1.97E+05|1.96E+05|1.96E+05|
> |AC|2.21E+03|2.21E+03|2.34E+03|2.21E+03|**2.20E+03**|
> |rds_cnt|1.49E+07|1.49E+07|1.50E+07|1.49E+07|1.49E+07|
> |HTRU2|8.21E+07|8.21E+07|8.61E+07|8.21E+07|8.21E+07|
> |GT|1.95E+07|1.95E+07|1.96E+07|1.95E+07|1.95E+07|
> |rds|476.88|476.88|486.75|476.88|**476.79**|
> |KEGG|4.94E+08|4.94E+08|4.95E+08|4.94E+08|4.94E+08|
> |urbanGB_10|1.15E+05|1.15E+05|1.26E+05|1.15E+05|1.15E+05|
> |rng_agr|8.23E+14|8.23E+14|-|8.23E+14|8.23E+14|
> |urbanGB|4.14E+05|4.14E+05|-|4.14E+05|4.14E+05|
> |spnet3D|2.28E+07|2.28E+07|-|2.28E+07|2.28E+07|
> |retail|6.80E+09|6.80E+09|-|6.80E+09|6.80E+09|
> |synthetic|9.44E+06|9.44E+06|-|9.44E+06|9.44E+06|
> |retail-II|2.90E+10|2.31E+10|-|2.31E+10|2.31E+10|
> |USO1990|6.91E+08|6.91E+08|-|6.91E+08|6.91E+08|
>
> * **Q3 - Q5.** Because of the space limit per comment, the response to these questions is provided in the next comment.

---

> > ### Author Response · Authors · 2022-08-02
> > **More Response to Reviewer FDa1**
> >
> > * **Q3 - Other than the centroid-based approaches, there are many other kinds of clustering, e.g. density-based, hierarchical clustering, etc. They often don't need the ball-shape assumption and work better in real world datasets. The paper lacks comparison with these different approaches, especially the state-of-the-art ones like [Sarfraz, et al, Efficient Parameter-free Clustering Using First Neighbor Relations].**
> >
> >     Different clustering methods have different formulations (models) with different objective functions and constraints. The performance of a clustering method is determined by its formulation and the quality of the solution algorithm. In our work, we focus on improving the quality of solutions for a fixed formulation (KMedoids). As we mentioned in response to Q1, we believe that the unique "medoids on samples" constraint of KMedoids is important for many applications. For applications where this constraint is not necessary, it is possible that other clustering formulations may lead to better performance, but finding a better formulation is out of the scope of this paper.
> >
> >     Moreover, as mentioned in response to *Q3 of Reviewer DaVW*, our method can potentially be extended to solve other clustering problems to global optimality. If we can solve all clustering methods to global optimality, then it enables a fair comparison between different clustering methods. Otherwise, it is hard to distinguish if the performance difference is caused by the formulations (e.g., KMedoids vs. Hierarchical clustering) or the solution algorithms (e.g., global optimal algorithm vs. heuristics). With the global optimum guaranteed, the community can concentrate more on developing more suitable formulations for realistic problems without the concern of lacking a high-quality solution.
> >
> > * **Q4 - In experiments, the performance are evaluated using the achieved optimal gap and number of solved nodes. Standard clustering metrics include ARI and NMI, which provide a sense of clustering quality. The gap is useful in theoretical settings but not very meaningful in applications.**
> >
> >     We thank the reviewer for the concern of ARI and NMI. In our original submission, we only compared the objective value (upper bound) and optimality gap. The objective value provides the best sum of distances within the cluster, and the optimality gap qualifies as the worst gap from the global optimal solution. We will add a discussion on ARI and NMI, and more numerical results, including the following table, in the final version of the paper.
> >
> >     ARI and NMI are standard indices to evaluate the clustering results with the ground truth. However, most datasets used in the paper do not have a ground truth (as an unsupervised learning method, clustering is often applied without knowing the ground truth for real applications), so it is impossible to compare them. Here, we performed additional numerical results and gave the ARI and NMI results of 5 datasets with ground truth labels in the following table. In these experiments, the number of clusters is set to the number of ground truth labels. From this table, our BB+LD method can give slightly better ARI and NMI on iris and HRTU2.
> >
> > | Dataset | Sample | Dimension | Cluster | Objective Value | Objective Value | ARI | ARI | NMI | NMI |
> > |:---:|:---:|:---:|:---:|:---:|:---:|:---:|:---:|:---:|:---:|
> > |  |  |  |  | Heuristic | BB+LD | Heuristic | BB+LD | Heuristic | BB+LD |
> > | iris | 150 | 4 | 3 | 84.68 | **83.96** | 0.73  | **0.75**  | 0.76  | **0.80**  |
> > | HCV | 572 | 12 | 4 | 2.2873E+06 | 2.2873E+06 | 0.44  | 0.44  | 0.25  | 0.25  |
> > | HF | 299 | 12 | 2 | 1.3512E+12 | 1.3512E+12 | 0.02  | 0.02  | 0.00  | 0.00  |
> > | HRTU2 | 17898 | 8 | 2 | 1.2536E+08 | **1.2535E+08** | -0.0780 | **-0.0779** | 0.0267 | **0.0271** |
> > | UK | 258 | 5 | 4 | 45.04 | 45.04 | 0.24  | 0.24  | 0.33  | 0.33  |
> >
> > * **Q5 - It would be better to introduce more on BB method in this setting, e.g. what are the BB tree and node? How are they constructed?**
> >
> >     Thank you for pointing out this issue. We will try to clarify the setting of the branch and bound procedure more clearly in the final version.

---

> > > ### Comment · Reviewer_FDa1 · 2022-08-06
> > > **Thanks but need more evidence**
> > >
> > > Thank you for providing additional experiments to address my concerns. I appreciate the effort obtain these tables. However, after checking the tables, I think more evidence is needed to justify the practical usage of the proposed method:
> > >
> > > 1. In the table that compares with K-Means (with projection centers to closest data points), the advantage in terms of objective value (sum of total distance) is not obvious, usually <= 1%. And most of the time they are just the same. Could you also provide the run time in this table, to show that if the proposed method has advantages in terms of the running speed?
> > >
> > > 2. In the second table:
> > > (1) On HF, HRTU2 and UK, an ARI value closer to 0, essentially means a random clustering. Therefore the results are very weak.
> > > (2) Could you also put the ARI and NMI values from K-Means, K-Means++? As well as the run times into this table?
> > > (3) There are some standard dataset with class labels, e.g. MNIST, REUTERS, that is often used to benchmark clustering. They also has more data samples, higher dimension and more number-of-clusters. Please consider testing on these middle or even large-scale dataset (e.g., ImageNet), to show the method can scale well.

---

> > > > ### Author Response · Authors · 2022-08-08
> > > > **More responce to Reviewer FDa1**
> > > >
> > > > Thank you very much for following up after our response.
> > > >
> > > > * **Q1 - In the table that compares with K-Means (with projection centers to closest data points), the advantage in terms of objective value (sum of total distance) is not obvious, usually <= 1%:** For the majority of the datasets, our algorithm converges to a solution with an optimal gap of 0.1%, which means that the solution we obtained is at most 0.1% worse than the global optimal solution. Here, global optimal solutions are of core consideration in many high-value-added fileds, such as inventory optimization, machine allocation, and office assignment. In these fields, with a proper model, **even little improvements (e.g., 1-2%) in the objective values can contribute large financial benefits**. Our method provides a guarantee of global optimum for the KMedoids problems, which can also be applied in many high-value-added fields, such as facility location, and chemical process anomaly detection. Moreover, the optimality gap obtained in our algorithm provides a baseline to evaluate the performance of a heuristic method. Only by comparing the heuristic solution with the global optimal solution as we did in the table, we can confidently claim that the heuristic method can do a fairly nice job in finding near-optimal solutions. For example, even after a user runs K-means a million times with random seeds, the user is still unsure if it is worth trying another run. The optimality gap of X% obtained from our algorithm qualifies as the worst gap from the global optimal solution. It thus helps the user decide whether to invest more time to get a solution that is at most X% better.
> > > >
> > > >     As for the run time, global optimal methods generally need significantly more time than heuristic methods, because heuristic methods do not need to guarantee optimality. So we compare the solution time with an off-the-shelf commercial global optimal solver - CPLEX. The results show that our method can solve larger problems (400 times larger) within the time limit. Moreover, in high-value-added fields, people generally care more about the quality of the solution as long as the run-time is acceptable (e.g., hours). Hence, a method with a global optimal guarantee is preferred.
> > > >
> > > > * **Q2 - Considerations about ARI and NMI results in Table 2:**
> > > > 1. **On HF, HRTU2 and UK, an ARI value closer to 0, essentially means a random clustering. Therefore the results are very weak:** ARI and NMI evaluate the performance of the formulation of a clustering method (in other words, how well the formulation matches the problem). However, the performance of a clustering method consists of two parts: the formulation and the quality of solutions. Our work focuses on improving the quality of solutions. It provides a global optimal solution to the K-Medoids formulation, which means our method can always obtain the best-qualified solution to the K-Medoids formulation. However, whether there is a better formulation than K-Medoids for the dataset is out of our consideration.
> > > >
> > > > 2. **ARI and NMI values from K-Means, K-Means++:**
> > > > ARI and NMI evaluate how well the formulation matches the true labels. Hence, a formulation's objective value may not be consistent with ARI and NMI. As seen from this table, PAM obtains the best ARI and NMI among the five methods for the HCV dataset, while it has the worst KMedoids objective value. Our method focuses on obtaining the best KMedoids objective values within an acceptable running time. However, if we need to compare with respect to ARI and NMI, our method always provides better or the same performance compared with K-means and K-means++.
> > > >
> > > > |  | Dataset | iris | HCV | HF | HRTU2 | UK |
> > > > |:---:|:---:|:---:|:---:|:---:|:---:|:---:|
> > > > |  | Sample | 150 | 572 | 299 | 17898 | 258 |
> > > > |  | Dimension | 4 | 12 | 12 | 8 | 5 |
> > > > |  | Cluster | 3 | 4 | 2 | 2 | 4 |
> > > > | UB | Kmeans | 84.68 | 2.2873E+06 | 1.3512E+12 | 1.2536E+08 | 45.04  |
> > > > | UB | Kmeans++ | 84.68 | 2.2873E+06 | 1.3512E+12 | 1.2536E+08 | 45.04  |
> > > > | UB | PAM | 91.04 | 2.3168E+06 | 1.3917E+12 | 1.2800E+08 | 45.42  |
> > > > | UB | Heuristic | 84.68 | 2.2873E+06 | 1.3512E+12 | 1.2536E+08 | 45.04 |
> > > > | UB | BB+LD | **83.96** | 2.2873E+06 | 1.3512E+12 | **1.2535E+08** | 45.04 |
> > > > | ARI | Kmeans | 0.73 | 0.44  | 0.02  | -0.0780  | 0.24  |
> > > > | ARI | Kmeans++ | 0.73 | 0.44  | 0.02  | -0.0780  | 0.24  |
> > > > | ARI | PAM | 0.70 | **0.59**  | 0.01  | **-0.0738**  | 0.15  |
> > > > | ARI | Heuristic | 0.73  | 0.44  | 0.02  | -0.0780  | 0.24  |
> > > > | ARI | BB+LD | **0.75**  | 0.44  | 0.02  | -0.0779  | 0.24  |
> > > > | NMI | Kmeans | 0.76  | 0.25  | 0.00  | 0.0267  | 0.33  |
> > > > | NMI | Kmeans++ | 0.76  | 0.25  | 0.00  | 0.0267  | 0.33  |
> > > > | NMI | PAM | 0.76  | **0.34**  | 0.00  | 0.0221  | 0.22  |
> > > > | NMI | Heuristic | 0.76  | 0.25  | 0.00  | 0.0267  | 0.33  |
> > > > | NMI | BB+LD | **0.80**  | 0.25  | 0.00  | **0.0271**  | 0.33  |

---

> > > > > ### Comment · Reviewer_FDa1 · 2022-08-10
> > > > > **Score update**
> > > > >
> > > > > Thanks for providing more numbers. Though the guaranteed optimality could be meaningful in certain cases, I am still questioning the practical usage of the proposed method in the task of clustering, due to the slow speed. As the paper successfully connect Cornuejols' work with BB method, I update my score to borderline accept.

---

### Official Review · Reviewer_htGe · 2022-07-11

**Rating:** 5
**Confidence:** 4
**Soundness:** 3 good
**Presentation:** 3 good
**Contribution:** 2 fair

**Summary:**

The K-Medoids, as a classic clustering method, have many real applications. To fix the pool scalability of classic branch and bound (BB), this work proposed a method with reduced-space BB, in which the lower bound is provided by Lagrangian relaxation. Combined with several tailored bound tightening techniques, the method was tested in 28 datasets in the experiments section, including two datasets with one million samples.

**Questions:**

It would be better if the author can emphasize the significance of their upper/lower bound method.

**Limitations:**

No potential negative societal impact.

**Strengths And Weaknesses:**

I do recognize the huge sets of experiments and the effort. Just saying, I have a feeling that: Compared to machine learning, this works probably more like the FOCS or STOC type of paper for theoretical computing.

Strengths：
The idea is intuitive and makes sense.

There is an experiments-side contribution since the author compares the state-of-art baselines. The author provided experiments on 28 datasets, including two datasets with one million samples. The experiment on this data size is impressive.

Several tailored bound tightening techniques needs effort as an engineering solution, especially when facing large-scale dataset.

Weaknesses：
Theoretical side, the focus should be on how we select the proper upper and lower bound in the BB. In this work, the upper bounding methods are fine but not surprising. And, as they mentioned, the tight lower bound from the work of Cornuejols.

---

> ### Author Response · Authors · 2022-08-02
> **Response to Reviewer htGe**
>
> Thank you for your review and valuable remarks!
>
> **Q1 - Theoretical side, the focus should be on how we select the proper upper and lower bound in the BB. In this work, the upper bounding methods are fine but not surprising. And, as they mentioned, the tight lower bound from the work of Cornuejols.**
>
> Thanks for the suggestion. We will add a discussion on the selection of bounds. Basically, they are selected to ensure global convergence and speed up the solution process.
>
> * **Selection of lower bound.**
> Our lower bound method consists of a basic lower bound and a Lagrangian-based lower bound. The Lagrangian-based lower bound is from Cornuejols et al. However, the Lagrangian method alone cannot close the duality gap, as shown in the numerical results. Even when it is combined with the reduced-space branch and bound (BB), convergence cannot be guaranteed. The basic lower bound plays an essential role because merely using the basic lower bound method can already guarantee the global convergence of the reduced-space BB. Moreover, since the basic lower bound has a closed-form solution, its computational cost is much cheaper than the Lagrangian-based lower bound. It can accelerate the computation by checking $\beta_{basic} > UB$. If so, this node can be deleted without calculating the Lagrangian-based lower bound. Besides, this basic lower bound is also utilized extensively in the bound tightening procedure to reduce the search space.
>
> * **Selection of upper bound.**
> We provide two efficient upper bounding methods. At the child node, we utilize the candidate solution obtained in the lower bounding method to update the upper bounds. This method guarantees the convergence of the upper bound and is computationally very fast.
>
>     At the root node, we use a combination of a K-Means-like method and Evolutionary Centers Algorithm (we call this combination HEURISTIC). We performed additional numerical experiments to illustrate that this method performs better than other popular heuristic methods in the literature, including Kmeans, Kmeans++, and PAM, on all 28 datasets. Because of the space limit per comment, detailed results are provided in the next comment.
>
> * **Significance.**
> Besides the significance of our lower/upper bound methods, we also would like to highlight the other novelty and contributions of this work.
>
>     From a theoretical perspective, we develop an efficient reduced-space BB scheme for the K-medoids clustering problem. Combined with the abovementioned lower/upper bounds, we prove the finite convergence by branching only on $A\times K$ variables, where $A$ is the number of features, and $K$ is the number of clusters. In contrast, the classical BB scheme needs to branch on all the $S^2$ binary variables, where $S$ is the number of samples. For example, the USO1990 dataset in Table 2 of the original submission has 2.5 million samples with 68 dimensions. In this case, if the cluster number $K=3$, the number of branching variables in our algorithm is $204$, while the classical BB scheme needs to branch on $6.25\times 10^{12}$ variables.
>
>     From the computational and experimental perspective, we propose several tailored bound tightening techniques to reduce the search space and computational cost significantly. Most importantly, we provide an open-source package that can address datasets with one million samples (400 times larger than the state-of-art work) and reach a small gap (0.1%) within one hour. To the best of our knowledge, no other article has ever reported results on datasets of this scale for K-Medoids problems with a global convergence guarantee.

---

> > ### Author Response · Authors · 2022-08-02
> > **More Response to Reviewer htGe**
> >
> > To illustrate the effectiveness of HEURISTIC, the following table compares its performance with several other popular heuristic methods in the literature, including Kmeans, Kmeans++, and PAM, on all 28 datasets. Here, we run all the methods many times with random seeds and select the best result. The centers of K-Means and K-Means++ are projected to the nearest samples to fulfill the "Medoids on Samples" constraint in the KMedoids problem. This table shows that the Heuristic can always obtain the same or better objective value than Kmeans, Kmeans++, and PAM.
> >
> > | dataset | Kmeans | Kmeans++ | PAM | HEURISTIC | BB+LD |
> > |:---:|:---:|:---:|:---:|:---:|:---:|
> > | iris | 84.63 | 84.63 | 90.99 | 84.63 | **83.91** |
> > | seeds | 598.29 | 598.29 | 608.72 | 598.29 | 598.29 |
> > | glass | 629.02 | 629.02 | 652.15 | 629.02 | 629.02 |
> > | BM | 8.65E+05 | 8.65E+05 | 9.17E+05 | 8.65E+05 | **8.63E+05** |
> > | UK | 50.77 | 51.19 | 51.06 | 50.77 | 50.77 |
> > | HF | 7.83E+11 | 7.83E+11 | 7.83E+11 | 7.83E+11 | 7.83E+11 |
> > | Who | 8.34E+10 | 8.34E+10 | 8.44E+10 | 8.34E+10 | **8.33E+10** |
> > | HCV | 2.85E+06 | 2.82E+06 | 2.76E+06 | 2.75E+06 | 2.75E+06 |
> > | Abs | 2.62E+06 | 2.62E+06 | 2.66E+06 | 2.62E+06 | 2.62E+06 |
> > | TR | 1.14E+03 | 1.14E+03 | 1.16E+03 | 1.14E+03 | **1.13E+03** |
> > | SGC | 1.28E+09 | 1.28E+09 | 1.49E+09 | 1.28E+09 | 1.28E+09 |
> > | hemi | 9.92E+06 | 9.91E+06 | 1.18E+07 | 9.91E+06 | 9.91E+06 |
> > | pr2392 | 2.13E+10 | 2.13E+10 | 2.53E+10 | 2.13E+10 | 2.13E+10 |
> > | TRR | 1.97E+05 | 1.96E+05 | 1.97E+05 | 1.96E+05 | 1.96E+05 |
> > | AC | 2.21E+03 | 2.21E+03 | 2.34E+03 | 2.21E+03 | **2.20E+03** |
> > | rds_cnt | 1.49E+07 | 1.49E+07 | 1.50E+07 | 1.49E+07 | 1.49E+07 |
> > | HTRU2 | 8.21E+07 | 8.21E+07 | 8.61E+07 | 8.21E+07 | 8.21E+07 |
> > | GT | 1.95E+07 | 1.95E+07 | 1.96E+07 | 1.95E+07 | 1.95E+07 |
> > | rds | 476.88 | 476.88 | 486.75 | 476.88 | **476.79** |
> > | KEGG | 4.94E+08 | 4.94E+08 | 4.95E+08 | 4.94E+08 | 4.94E+08 |
> > | urbanGB_10 | 1.15E+05 | 1.15E+05 | 1.26E+05 | 1.15E+05 | 1.15E+05 |
> > | rng_agr | 8.23E+14 | 8.23E+14 | - | 8.23E+14 | 8.23E+14 |
> > | urbanGB | 4.14E+05 | 4.14E+05 | - | 4.14E+05 | 4.14E+05 |
> > | spnet3D | 2.28E+07 | 2.28E+07 | - | 2.28E+07 | 2.28E+07 |
> > | retail | 6.80E+09 | 6.80E+09 | - | 6.80E+09 | 6.80E+09 |
> > | synthetic | 9.44E+06 | 9.44E+06 | - | 9.44E+06 | 9.44E+06 |
> > | retail-II | 2.90E+10 | 2.31E+10 | - | 2.31E+10 | 2.31E+10 |
> > | USO1990 | 6.91E+08 | 6.91E+08 | - | 6.91E+08 | 6.91E+08 |

---

> > ### Comment · Reviewer_htGe · 2022-08-07
> > **Response to the author**
> >
> > I thank the author for the response.
> > On the one hand, I appreciate the experiment-side effort and contribution, as I claimed in my review.
> > I agree with you about the novelty of your reduced-space BB scheme on the large-scale K-medoids clustering problem.
> > This part is excellent, and I like it.
> >
> > On the other hand, the 'significance of our lower/upper bound methods' is my main concern. I do understand your response to those selections.
> > Could you please clarify the following thing about your lower/upper bound methods?
> > 1. How do you evaluate the contribution of your basic LB with respect to the theoretical hardness?
> > 2. The upper bound is not considered as your significant contribution. Please correct me if you have your novelty for the upper bound.
> >
> > Honestly, this is a hard decision since I have two opposite opinions on the two sides I mentioned. I will hold a positive opinion if you address my main concern.

---

> > > ### Author Response · Authors · 2022-08-08
> > > **More response to Reviewer htGe**
> > >
> > > Thank you for your further response! The following are our explanations regards to your main concerns. We hope that through the rebuttal and discussion, we will resolve the reviewer's concerns and convince the reviewer to raise their evaluation. We would really appreciate the support!
> > >
> > > * **Q1 - How do you evaluate the contribution of your basic LB with respect to the theoretical hardness?**
> > >
> > >     The basic LB relaxes two constraints: (1) the nonanticipativity constraint: all the samples share the same medoid set; (2) the "medoids on samples" constraint. We acknowledge that this basic LB is intuitive. However, despite its simple form, it is theoretically essential to guarantee the convergence of the algorithm and computationally very effective.
> > >
> > >     Theoretically, one of our key contributions is that we proved that the convergence of the reduced-space branch and bound (BB) could be guaranteed using the proposed lower and upper bounds. However, since the reduced-space BB only branches on a very small subset of variables, its convergence can only be guaranteed when combined with suitable lower and upper bounding methods. For example, linear relaxation (removing the integrality constraint or relaxing a binary variable to be a continuous variable between 0 and 1) is a very popular method to get LB in the mixed-integer optimization community and is implemented in several state-of-the-art solvers such as CPLEX and Gurobi. However, the reduced-space BB with this lower bound can not converge. In particular, in the extreme case of branching only on the space of medoids, even if the medoids are fixed, the LB generated by the linear relaxation will not converge. Surprisingly, merely using the basic lower bound method can already guarantee the global convergence of the lower bound of the reduced-space BB.
> > >
> > >     Computationally, we obtain a closed-form solution of the basic LB with a complexity of $O(S)$ (Lagrangian LB needs $O(S^2)$). Since the basic LB is computationally much cheaper, we can accelerate the overall solving process by checking $\beta_{basic} > UB$. If so, this node can be deleted without computations of Lagrangian LB. For the million-scale datasets, this checking deleted 45 nodes in a total of 214 nodes in Retail-II. Moreover, the basic LB is also utilized in the feasibility check of FBBT, where Lagrangian LB is unsuitable. For the million-scale datasets, FBBT reduced 2.45e7 sample counts in a total of 4.11e7 sample counts in Retail-II (Here, sample count means the sum of sample numbers in the medoid regions for all the LB iterations); FBBT reduced 5.67e5 sample counts in a total of 1.16e6 sample counts in USO1990.
> > >
> > >
> > >
> > > * **Q2 - The upper bound is not considered as your significant contribution. Please correct me if you have your novelty for the upper bound.**
> > >
> > >     At the child node, we utilize the candidate solution obtained in the lower bounding method to update the upper bounds. Again, despite its simple form, the upper bound is theoretically essential to guarantee the convergence of the upper bound of the reduced-space BB. We believe that a simple form with a strong theoretical guarantee is a merit of our algorithm.
> > >
> > >     At the root node, the upper bound consists of the K-Means-like method and ECA. While we only combine existing methods here, it can provide better or equal initial UB guesses than any of them.
> > >
> > >     Besides LB and UB, it should be noted that we also provide several efficient bound tightening methods, which enable our algorithm to address large-scale datasets.

---

> > > > ### Comment · Reviewer_htGe · 2022-08-09
> > > > **Response to the author**
> > > >
> > > > I thank the author for the detailed explanation.
> > > > I agree that the elegance of the simple form and the techniques for experiments on large-scale data.
> > > > However, I hold my opinion on the novelty and significance of bounding methods.
> > > >
> > > > I will leave the judgment to the AC and other reviewers.

---

> > > > ### Comment · Reviewer_htGe · 2022-08-09
> > > > **Score update**
> > > >
> > > > I thank the author for the experiment-side efforts and detailed explanation.
> > > > Even though I hold my opinion on the bounding method part, I slightly raise my score for the elegance of the simple form and the techniques for experiments on large-scale data.

---

### Official Review · Reviewer_2o3M · 2022-07-12

**Rating:** 7
**Confidence:** 4
**Soundness:** 4 excellent
**Presentation:** 3 good
**Contribution:** 3 good

**Summary:**

The paper presents a spatial branch-and-bound method for the K-medoids clustering problem. The authors introduce several bound tightening techniques tailored for their approach. In numerical experiments they demonstrate that their method is able to solve large scale instances of the problem to global optimality.

**Questions:**

The presentation of the results is mostly clear, but I have a few questions/comments:

In lines 96-97 you are mentioning that the value $d_{s,j}$ are precomputed. However, the number of values scales quadratically with the size of  the data set, which seems intractable for the claimed size of 1M samples. How do you deal with this issue?

In line 130, the authors define $S^{k+}$ as the set of indexes whose samples are contained in the $k$-th medoid constraint set. Then, in equation (9) we consider the union $S^+ = S^{1+} \cup \dotso \cup S^{K+}$, which should just be the set of all sample indexes. Can you please make your notation more clear as to what you actually mean here?

In line 166 please clarify the notation $\mathrm{mid}$.

Typos/vocabulary:
- line 83: it should be $\mathbb R^{S \times A}$
- line 85: "objective"
- line 124: "belongings" -> "assignment"
- line 227: "miscellaneous"
- line 228: "symmetry-breaking"

**Strengths And Weaknesses:**

The paper makes a strong technical contribution to a computationally challenging problem. The writing is clear and concise, the numerical results are significant. Overall it's a good paper with only few flaws.

---

> ### Author Response · Authors · 2022-08-02
> **Response to Reviewer 2o3M**
>
> We sincerely thank the reviewer for their positive and insightful comments and suggestions. We will respond to your questions in order.
>
> * **Q1 - In lines 96-97 you are mentioning that the value $d_{s,j}$ are precomputed. However, the number of values scales quadratically with the size of the data set, which seems intractable for the claimed size of 1M samples. How do you deal with this issue?** Yes, the memory consumption of precomputed $d_{s,j}$ is $O(S^2)$, where $S$ is the number of samples. For example, a dataset with 100,000 samples will need 74.5GB RAM to store $d_{s,j}$ in the Float64 format, while a dataset with 1,000,000 will need 7,450 GB RAM.
>
>     In our implementation, the datasets with no more than 100,000 samples were computed on one compute node with 40 cores and 202GB RAM.
>
>     For datasets with more than 100,000 samples, we executed the experiments on multiple compute nodes. Each core precomputes and stores part of the $d_{s,j}$ matrix, as described in the revised *Parallel implementation section in Q2*. Then, each process computes the corresponding contributions using the portion of $d_{s,j}$ matrix stored in this process. Finally, the contributions from each process are combined to generate the final results. For example, we executed the one-million dataset RETAIL-II on 150 nodes with 6,000 cores and 30,300 GB of RAM. In this way, each core needs to store 1.25GB of $d_{s,j}$.
>
>     For datasets with more than 1,000,000 samples, we calculated $d_{s,j}$ on the fly, without precomputing and storing $d_{s,j}$. In this case, the complexity of one Lagrangian update iteration increases from $O(S^2)$ (if $d_{s,j}$ are precomputed) to $O(AS^2)$ (if not), where $A$ is the number of features. Hence, we can expect an acceptable slowdown when the dimensions of datasets are small. For example, in Table 2 of the original submission, the dataset USO1990 with two-million samples, and 68 features was executed without precomputing $d_{s,j}$, and the result is acceptable. We will note it in the revised version.
>
>     We also performed additional numerical experiments on several datasets to compare the performance of precomputing $d_{s,j}$ and calculating on the fly. The following table shows that although calculating $d_{s,j}$ on the fly is slower than precomputing $d_{s,j}$, the slowdown is acceptable. We will add the complete comparison in our final version.
>
> | Dataset | Dimension | Total Run Time(s) | Total Run Time(s) | Time per LD Iteration(ms) | Time per LD Iteration(ms) |
> |:---:|:---:|:---:|:---:|:---:|:---:|
> |  |  | on the fly | $d_{s,j}$ pre-computed | on the fly | $d_{s,j}$ pre-computed |
> | Abs | 21 | 34 | 12 | 1.42 | 0.70 |
> | hemi | 7 | 17 | 13 | 9.91 | 2.02 |
> | rds_cnt | 4 | 47 | 30 | 175 | 37.80 |
> | TR | 10 | 58 | 15 | 2.54 | 0.73 |
> | TRR | 24 | 365 | 74 | 110 | 10.80 |
>
> * **Q2 - In line 130, ..., Can you please make your notation more clear as to what you actually mean here?** Thank you for pointing out these confusing notations. We will change the description to make it more clear in the final version:
>     > **Parallel implementation:** we implement a parallel version of the tailored Lagrangian relaxation, in which the computations of contributions, $\rho_j(\lambda)$, are performed in each process. At the beginning of the BB procedure, given $P$ processes, the index set $\mathcal{S}$ of the whole dataset is evenly divided into $P$ subsets, which is $\mathcal{S}\_p, p=1,\cdots,P$. Then, each process computes and stores the distance assigned to it, which is $d_{s,j}, s\in\mathcal{S}, j \in\mathcal{S}_p$.
>    >
>    > At each BB node with $M$, denote the feasible index set of all the medoids as $$\mathcal{S}^+(M) = \mathcal{S}^{1+}(M) \cup \mathcal{S}^{2+}(M) \cup \cdots \cup \mathcal{S}^{K+}(M).$$
>    >
>    > Then, each process use the stored portion of $d_{s,j}$ to computes the assigned contributions of $\rho_{j}(\lambda), \forall j \in \mathcal{S}_p\cap \mathcal{S}^+(M)$.
>    >
>    > The remaining parts of the parallel implementation (e.g., update of $\lambda$) are identical to the serial version.
>
> * **Q3 - In line 166 please clarify the notation mid.** *mid* here means the median value. We will add the notation in the final version.
>
> * **Q4 - Typos/vocabulary.** Thank you very much for your careful review and for catching these mistakes. We will fix the typos in the final version.

---

### Official Review · Reviewer_DaVW · 2022-07-14

**Rating:** 7
**Confidence:** 4
**Soundness:** 4 excellent
**Presentation:** 4 excellent
**Contribution:** 3 good

**Summary:**


The authors propose a branch and bound approach to solving the K-mediod problem by taking advantage of the Lagrangian relaxation method explored by Cornuejols in
 the 70's. In particular, they modify the lagrangian relaxation for each cluster by building appropriate boxes for the BB scheme and provide various speedups by
probing, feasibility based bound tightening and parallelization. Experiments are shown on a wide variety of datasets.


**Questions:**

(A) How does the scaling look like in terms of the dimensions d? Also, in general what is the computational complexity of the proposed algorithm? Of course, it w
ould depend on the number of the branch and bound steps.

(B) The algorithm seem to be more sensitive to dimensions than to samples in terms of performance (Table 2). Is that really true and if so, any reason why that could be the case?


**Limitations:**

The key reason it works is because of the Lagrangian relaxation available to the K-mediod setting and it is not clear if this will extend to other clustering problems. Can the authors shed some light as to where else this could be applicable?

**Strengths And Weaknesses:**

The paper is clearly written with references to related work and is easy to follow. The theoretical results are shown in the main paper with the proofs delegated
 to the supplementary. Additional experiments with varying sizes of clusters are also provided.

The problem is of broad interest since clustering is a core component of many machine learning tasks. The key ideas of using Lagrangian relaxtion and branch and
bound techniques have been explored in the broader literature but combining them in the context K-mediods problem along with mediods on samples constraint makes
it novel and relevant.

The paper pretty much builds on the work of Cornujols et al as well as the branch/bound literature and the proofs of finite termination/global optimum are provided.  Most of the literature for the relevant work is covered and referenced in the paper.

---

> ### Author Response · Authors · 2022-08-02
> **Response to Reviewer DaVW**
>
> Thank you very much for your encouraging comments and insightful feedback. We provide clarifications and answers to your questions and comments below.
>
> * **Q1 - How does the scaling look like in terms of the dimensions d? Also, in general what is the computational complexity of the proposed algorithm?**
>
>     In our algorithm, the majority of the runtime is spent on the computation of the Lagrangian-based lower bound. Denoting the number of samples as $S$, the number of features as $A$, and the number of clusters as $K$, the computational complexity of one Lagrangian update iteration is $O(S^2)$ (if $d_{s,j}$ are precomputed as shown in our paper) or $O(AS^2)$ (if $d_{s,j}$ are not precomputed). It should be noted that the number of branch and bound (BB) nodes is hard to predict and is not considered here. Also, the number of Lagrangian update iterations in each BB node is different because we set dynamic stopping criteria in case there is no update of LB in several continuous Lagrangian iterations.
>
>     Here, We performed additional numerical experiments on synthetic datasets to illustrate the complexity. These synthetic datasets were generated with 3 Gaussian clusters. We performed experiments on datasets with different numbers of samples and dimensions. $d_{s, j}$ are precomputed at the beginning of the branch and bound procedure. As shown in the following table, the average runtime of one Lagrangian iteration remains almost the same as the dimension changes. Moreover, the average runtime increases almost quadratically as the number of samples increases. These are consistent with the complexity mentioned above.
>
>     We will incorporate the reviewer's suggestion and add a discussion on the complexity in the revised version.
>
> * **Q2 - The algorithm seem to be more sensitive to dimensions than to samples in terms of performance (Table 2). Is that really true and if so, any reason why that could be the case?**
>
>     As mentioned in Q1, the complexity of one Lagrangian iteration is irrelative to the dimension $A$ if $d_{s,j}$ are precomputed as shown in our paper. The main reason for dimension sensitivity is the number of branching variables, which may affect the number of BB nodes. In our algorithm, we only need to branch on the regions of medoids, which consist of $A\times K$ variables (In contrast, the classical BB scheme needs to branch on all the $S^2$ binary variables, the dimension of which is typically much larger). Although the number of branching variables is $A\times K$, as shown in the following table, the actual number of explored BB nodes does not strictly follow the growth of dimensions. Also, the introduction of bound tightening methods makes the analysis more complicated.
>
>
> | Samples | Dimension | 2 | 5 | 10 | 20 | 30 | 40 | 50 | 60 | 70 | 80 | 90 | 100 |
> |:---:|:---:|:---:|:---:|:---:|:---:|:---:|:---:|:---:|:---:|:---:|:---:|:---:|:---:|
> | 1,000 | Nodes | 10 | 3 | 3 | 31 | 19 | 3 | 3 | 39 | 7 | 3 | 69 | 3 |
> |  | Time per LD Iteration (ms) | 4.28 | 4.28 | 4.06 | 3.57 | 3.62 | 4.02 | 4.11 | 3.78 | 4.32 | 3.8 | 3.21 | 3.79 |
> | 10,000 | Nodes | 3 | 17 | 7 | 17 | 17 | 15 | 41 | 13 | 7 | 3 | 201 | 17 |
> |  | Time per LD Iteration (ms) | 423 | 363 | 387 | 357 | 390 | 361 | 373 | 366 | 369 | 363 | 211 | 358 |
> | 100,000 | Nodes | 4 | 7 | 27 | 31 | 3 | 15 | 27 | 13 | 25 | 7 | 25 | 21 |
> |  | Time per LD Iteration (s) | 40.6 | 39.2 | 34.3 | 33.4 | 35.2 | 28.7 | 35.6 | 37.1 | 34.6 | 36.2 | 35.2 | 34.3 |
>
> * **Q3 - The key reason it works is because of the Lagrangian relaxation available to the K-mediod setting and it is not clear if this will extend to other clustering problems. Can the authors shed some light as to where else this could be applicable?:**
>
>     Our lower bounding method consists of the basic lower bound and Lagrangian-based lower bound. First, the basic lower bound method can be easily extended to many other clustering problems (e.g., hierarchical clustering) since it only needs to relax the implicit constraint that all the samples share the same cluster center points. Merely using the basic lower bound method can already guarantee the global convergence of the reduced space BB and its computational cost per node is very low (since we have derived its closed-form solution). Second, we believe that the Lagrangian relaxation method can also be extended to other clustering problems by dualizing a different set of constraints, as long as the objective function is to minimize the average cost over samples (instead of worst-case cost). However, the K-medoids problem is the only application in which the Lagrangian duality gap can be guaranteed within a certain threshold. So for other clustering problems, numerical experiments are needed to validate the tightness of the Lagrangian-based lower bound.
>
>     We will add extending to other problems as interesting directions for future work.

---

### Meta-Review · Area_Chair_SHCC · 2022-08-25

**Recommendation:** Accept
**Confidence:** Certain

**Metareview:**

The authors solve an important theoretical problem via a nice connection to Lagrangian relaxation for k-medoids and k-medoids on samples. The paper also has significant experimental results. The paper is also well-written, and information from the rebuttal should be incorporated into the final version. I recommend acceptance.


**Award:**

No

---

### Decision · Program_Chairs · 2022-09-14

Accept